# Dynamics at the serine loop underlie differential affinity of cryptochromes for CLOCK:BMAL1 to control circadian timing

Jennifer L Fribourgh[1†], Ashutosh Srivastava[2†], Colby R Sandate[3†], Alicia K Michael[1], Peter L Hsu[4], Christin Rakers[5], Leslee T Nguyen[1], Megan R Torgrimson[1], Gian Carlo G Parico[1], Sarvind Tripathi[1], Ning Zheng[4,6], Gabriel C Lander[3], Tsuyoshi Hirota[2], Florence Tama[2,7,8]*, Carrie L Partch[1,9]*

[1]Department of Chemistry and Biochemistry, University of California Santa Cruz, Santa Cruz, United States; [2]Institute of Transformative Bio-Molecules, Nagoya University, Nagoya, Japan; [3]The Scripps Research Institute, La Jolla, United States; [4]Department of Pharmacology, University of Washington, Seattle, United States; [5]Graduate School of Pharmaceutical Sciences, Kyoto University, Kyoto, Japan; [6]Howard Hughes Medical Institute, Seattle, United States; [7]Department of Physics, Nagoya University, Nagoya, Japan; [8]RIKEN Center for Computational Science, Kobe, Japan; [9]Center for Circadian Biology, University of California San Diego, La Jolla, United States

*For correspondence:
florence.tama@nagoya-u.jp (FT);
cpartch@ucsc.edu (CLP)

†These authors contributed equally to this work

Competing interests: The authors declare that no competing interests exist.

**Abstract** Mammalian circadian rhythms are generated by a transcription-based feedback loop in which CLOCK:BMAL1 drives transcription of its repressors (PER1/2, CRY1/2), which ultimately interact with CLOCK:BMAL1 to close the feedback loop with ~24 hr periodicity. Here we pinpoint a key difference between CRY1 and CRY2 that underlies their differential strengths as transcriptional repressors. Both cryptochromes bind the BMAL1 transactivation domain similarly to sequester it from coactivators and repress CLOCK:BMAL1 activity. However, we find that CRY1 is recruited with much higher affinity to the PAS domain core of CLOCK:BMAL1, allowing it to serve as a stronger repressor that lengthens circadian period. We discovered a dynamic serine-rich loop adjacent to the secondary pocket in the photolyase homology region (PHR) domain that regulates differential binding of cryptochromes to the PAS domain core of CLOCK:BMAL1. Notably, binding of the co-repressor PER2 remodels the serine loop of CRY2, making it more CRY1-like and enhancing its affinity for CLOCK:BMAL1.

## Introduction

The circadian clock links our behavior and physiology to the daily light-dark cycle, providing a timekeeping system that ensures cellular processes are performed at an optimal time of day. At the cellular level, circadian rhythms are driven by a set of interlocked transcription-based feedback loops that take ~24 hr to complete. The basic helix-loop-helix PER-ARNT-SIM (bHLH-PAS) domain-containing transcription factor CLOCK:BMAL1 forms the positive component of the core feedback loop (*Gekakis et al., 1998*). Several CLOCK:BMAL1 target genes (*Per1*, *Per2*, *Cry1*, *Cry2*) constitute the repressive components that close the core feedback loop (*Takahashi, 2017*). PER and CRY proteins interact with each other to form a large complex with the kinase CK1δ and enter the nucleus after a delay to bind directly to CLOCK:BMAL1 and inhibit its activity (*Aryal et al., 2017*; *Lee et al., 2001*). An additional feedback loop comprising the ROR and REV-ERB nuclear receptors controls the rhythmic expression of a subset of genes, including *Bmal1* (*Preitner et al., 2002*).

Cryptochromes are essential for circadian rhythms, as $Cry1^{-/-}$; $Cry2^{-/-}$ mice are arrhythmic in constant darkness (*van der Horst et al., 1999*; *Vitaterna et al., 1999*). Despite their high sequence and structural similarity (*Michael et al., 2017b*), CRY1 and CRY2 appear to have distinct roles in the molecular circadian clock because $Cry1^{-/-}$ mice have a short period, while $Cry2^{-/-}$ mice have a long period (*van der Horst et al., 1999*; *Vitaterna et al., 1999*). Using a genetic reconstitution assay of $Cry1^{-/-}$; $Cry2^{-/-}$; $Per2^{Luc}$ fibroblasts, we previously showed that tuning CRY1 affinity for CLOCK: BMAL1 can control circadian period (*Gustafson et al., 2017*; *Xu et al., 2015*); notably, the period lengthens when CRY1 binds more tightly to CLOCK:BMAL1, suggesting that the strength of CRY repression contributes to period determination. Consistent with this, CRY1 is a stronger repressor of CLOCK:BMAL1 than CRY2 (*Griffin et al., 1999*) and a recently discovered allele of CRY1 that enhances its repressive function also lengthens circadian period (*Patke et al., 2017*) in humans, providing a conceptual framework to understand why the presence of one CRY or the other influences circadian period.

Another notable difference between the two cryptochromes is the delay in expression of *Cry1* with respect to *Cry2* and the *Per* genes in the core feedback loop (*Lee et al., 2001*; *Ukai-Tadenuma et al., 2011*). The delayed expression of CRY1 is consistent with its recruitment to DNA-bound CLOCK:BMAL1 in two distinct phases: a minor peak at circadian time (CT) 16–20 as part of the large PER-CRY repressive complexes (*Aryal et al., 2017*; *Lee et al., 2001*), and a major peak later at CT0-4 that is independent of CRY2 and the PER proteins (*Koike et al., 2012*). However, the delayed timing of CRY1 expression does not account for its differential regulation of circadian period, because expressing *Cry2* from a minimal *Cry1* promoter that recapitulates this delay in $Cry1^{-/-}$; $Cry2^{-/-}$; $Per2^{Luc}$ fibroblasts or the suprachiasmatic nucleus (SCN) ex vivo still drives CRY2-like short periods (*Edwards et al., 2016*; *Rosensweig et al., 2018*). These data demonstrate that the differences between CRY1 and CRY2 that influence circadian period are encoded by their protein structure and/or dynamics to influence their repressive function.

What structural features of CRY1 and CRY2 might lead to differential functions in the circadian clock? Both cryptochromes are defined by a highly conserved photolyase homology region (PHR) domain and divergent C-terminal tails that are intrinsically disordered (*Partch et al., 2005*). Deletion or post-translational modification of the unstructured tails can modulate rhythm amplitude and period length (*Gao et al., 2013*; *Li et al., 2016*; *Liu and Zhang, 2016*; *Patke et al., 2017*), but the PHR domain is required to generate circadian rhythmicity (*Khan et al., 2012*). We recently showed that CRY1 interacts directly with both CLOCK and BMAL1 through two distinct regions on its PHR domain (*Michael et al., 2017a*; *Xu et al., 2015*; *Figure 1A*). The CC helix binds directly to the transactivation domain (TAD) of BMAL1 (*Figure 1B*), sequestering it from coactivators to directly repress CLOCK:BMAL1 activity (*Czarna et al., 2011*; *Xu et al., 2015*). However, repression also relies on the recruitment of CRY1 to the PAS domain core of CLOCK:BMAL1 via the CLOCK PAS-B domain, which docks into the evolutionarily conserved secondary pocket on the CRY1 PHR domain (*Michael et al., 2017a*).

Here we leverage these insights to identify a key biochemical feature that distinguishes the different repressive capabilities of CRY1 and CRY2. We found that changes in the structure and dynamics of the serine loop, located adjacent to the secondary pocket, allow CRY1 to bind the PAS domain core of CLOCK:BMAL1 with significantly higher affinity than CRY2. Moreover, substitutions in the serine loop and secondary pocket between CRY1 and CRY2 that influence circadian period (*Rosensweig et al., 2018*) are sufficient to modulate affinity of the CRY PHR domain for CLOCK: BMAL1. Finally, we found that the CRY-binding domain (CBD) of PER2 differentially remodels the structure of the serine loop of CRY1 and CRY2 to help equalize their affinity for the PAS domain core of CLOCK:BMAL1. These data provide a biochemical rationale linking the repressive function of CRY2 to the PER proteins, as well as explaining how CRY1 can act as a repressor of CLOCK:BMAL1 in the presence or absence of PER proteins.

## Results

### CRY1 and CRY2 bind similarly to the transactivation domain of BMAL1

To begin to understand how CRY1 and CRY2 differ as circadian repressors, we first compared their affinity for the BMAL1 TAD, as sequestration of the TAD from coactivators represents a direct

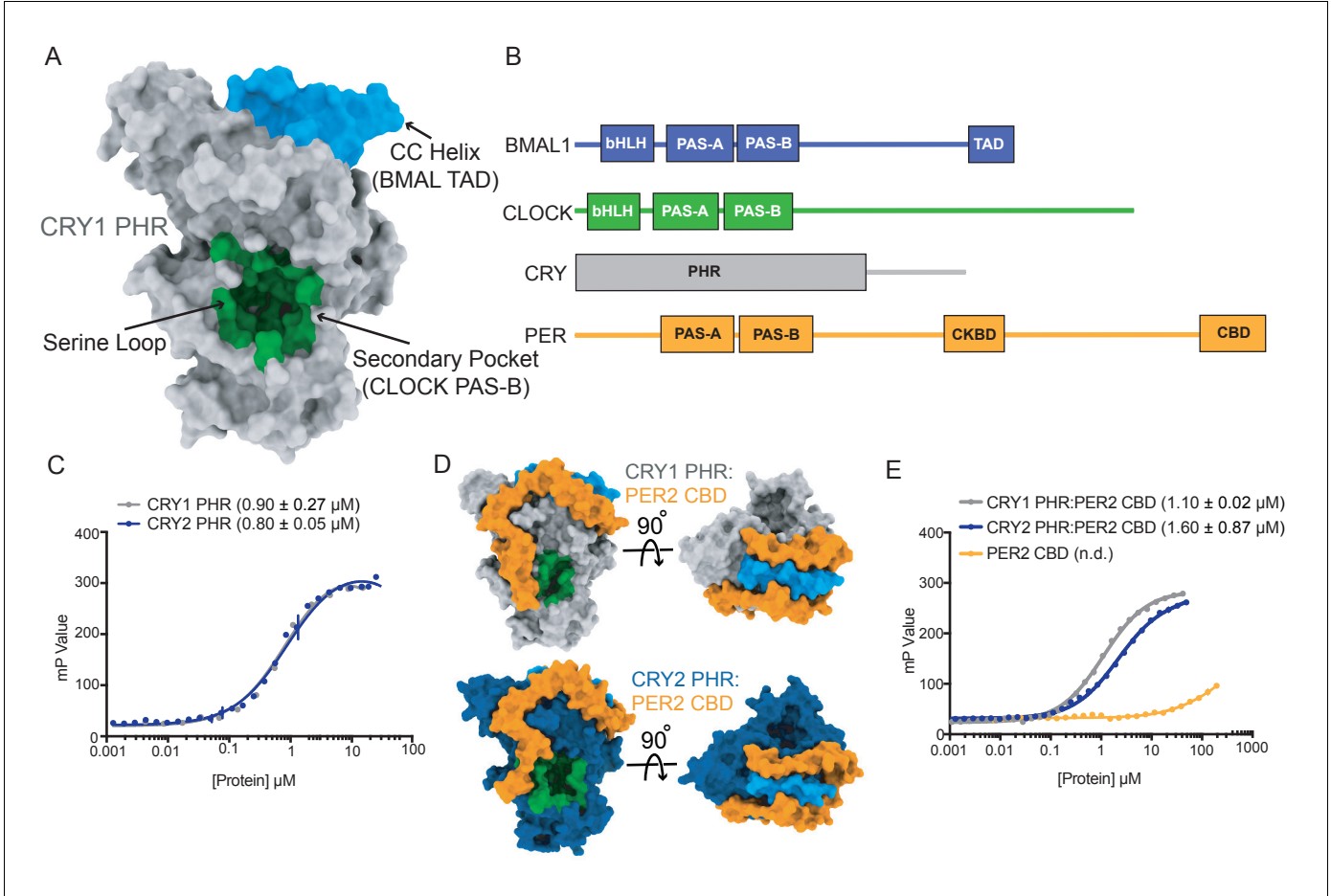

**Figure 1.** CRY1 and CRY2 bind similarly to the BMAL1 transactivation domain. (**A**) Crystal structure of the CRY1 PHR domain (PDB: 5T5X) highlighting the CC helix (BMAL1/2 TAD binding interface (blue)) and the secondary pocket (CLOCK PAS-B binding interface (green)). This is the standard orientation of the CRY1 and CRY2 PHR domain used as a reference for all other views. (**B**) Domain schematic of core clock proteins. bHLH, basic helix-loop-helix; PAS, PER-ARNT-SIM domain, TAD, transactivation domain; PHR, photolyase homology region; CKBD, Casein Kinase Binding Domain; CBD, CRY-binding domain. (**C**) Fluorescence polarization (FP) assay of 5,6-TAMRA-labeled BMAL1 minimal TAD (residues 594–626) binding to CRY1 PHR (gray), replotted from *Gustafson et al. (2017)*, and CRY2 PHR (blue). Mean ± SD data shown from one representative assay of n = 3 independent assays. Binding constants (mean ± SD) derived from n = 3 assays. (**D**) Crystal structures of CRY1 PHR:PER2 CBD (PDB: 4CT0) and CRY2 PHR:PER2 CBD (PDB: 4U8H) with the PER2 CBD (orange), CRY1 PHR (gray) and CRY2 PHR (blue), in the standard view and 90° rotated to show the BMAL1 TAD binding interface (blue). (**E**) FP assay of 5,6-TAMRA-labeled BMAL1 TAD binding to preformed CRY1 PHR:PER2 CBD (gray), CRY2 PHR:PER2 CBD (blue) or PER2 CBD (orange). Mean ± SD data shown from one representative assay of n = 3 independent assays. Binding constants shown (mean ± SD) were derived from n = 3 assays. n.d., not determined.

mechanism by which cryptochromes inhibit transcriptional activation by CLOCK:BMAL1 (*Gustafson et al., 2017*; *Xu et al., 2015*). We used a fluorescence polarization (FP)-based binding assay with a TAMRA-labeled BMAL1 TAD probe to demonstrate that CRY1 and CRY2 PHR domains both bind to the TAD with a similar, low micromolar affinity (*Figure 1C*). However, the repressive clock protein complexes that accumulate early in the night contain both CRYs with the associated co-repressor PER proteins (*Aryal et al., 2017*). Crystal structures of CRY1 and CRY2 PHR domains in their PER2-bound state (*Nangle et al., 2014*; *Schmalen et al., 2014*) reveal that the PER2 CBD wraps around the CC helix of both CRYs (*Figure 1D*), coming into close proximity of the BMAL1 TAD binding site (*Czarna et al., 2011*; *Xu et al., 2015*). To determine if PER2 binding influences the affinity of CRY1 or CRY2 for the BMAL1 TAD, we assessed binding of preformed CRY PHR:PER2 CBD complexes or the PER2 CBD alone to the TAMRA-BMAL1 TAD probe. Despite the intimate association of PER2 with the BMAL-binding site on CRY1 and CRY2, we found that both CRY:PER2 complexes bound the TAD similarly to the free CRYs, while the PER2 CBD displayed negligible

affity for the TAD (*Figure 1E*). These data establish that both CRY proteins have a similar capacity to sequester the TAD, either independently or in complex with the PER2 CBD, to repress CLOCK: BMAL1 activity (*Xu et al., 2015*).

## CRY1 binds substantially tighter than CRY2 to the PAS domain core of CLOCK:BMAL1

We next focused our attention on the PAS domain core of CLOCK:BMAL1, as we previously showed that the PAS-B domain of CLOCK docks into an evolutionarily conserved secondary pocket on CRY1 (*Michael et al., 2017a*). Importantly, binding of CRY1 to these two distinct sites on CLOCK and BMAL1 is important for repression, as disruption of either interface hinders repression by CRY1, while disruption of both interfaces eliminates the ability of CRY1 to act as a repressor for CLOCK: BMAL1 (*Sato et al., 2006*; *Xu et al., 2015*).

We initially monitored formation of stable CRY PHR-CLOCK:BMAL1 complexes by gel filtration chromatography using a construct of CLOCK:BMAL1 that contains the structured basic helix-loop-helix DNA-binding domain and tandem PAS domains (bHLH-PAS) that form the structured core of the heterodimer (*Huang et al., 2012*). Upon mixing the bHLH-PAS heterodimer with an excess of either CRY1 or CRY2 PHR domain, we observed that only CRY1 could bind CLOCK:BMAL1 tightly enough to co-migrate on the column (*Figure 2A,B*, *Figure 2—figure supplement 1*), suggesting that there was a substantial difference in affinity of the CRY PHR domains for the PAS domain core of CLOCK:BMAL1. To quantitatively analyze this difference in affinity, we performed bio-layer interferometry (BLI) with the biotinylated tandem PAS-AB domain core of the heterodimer, titrating in free CRY1 or CRY2 PHR domain to assess differences in CRY binding (*Figure 2C,D*). We found that the CRY1 PHR domain binds with a remarkably high affinity ($K_d$65 ± 6 nM), while CRY2 bound with approximately 20-fold lower affinity ($K_d$1.2 ± 0.2 μM).

The low micromolar affinity of CRY2 for the CLOCK:BMAL1 PAS domain core likely explains its inability to form a stable complex with CLOCK:BMAL1 under the conditions used in this assay. To see if reconstituting the multivalent interactions with CLOCK and BMAL1 could stabilize CRY2 binding to CLOCK:BMAL1, we assayed for stable complex formation in a sample containing the CRY2 PHR domain with a heterodimer containing the truncated CLOCK bHLH-PAS domain and full-length BMAL1 by gel filtration. In contrast to the isolated bHLH-PAS core, the CRY2 PHR domain now co-eluted with the CLOCK:BMAL1 heterodimer, as visualized by SDS-PAGE of the peak fraction (*Figure 2—figure supplement 1*). Therefore, although CRY2 binds with much lower affinity to the PAS core of CLOCK:BMAL1 than CRY1, the addition of a second binding interface with the BMAL1 TAD facilitates stable complex formation with the CRY2 PHR domain. Together, these data suggest that differential affinity of CRY PHR domains for the CLOCK:BMAL1 PAS domain core might underlie their distinctive repressive roles in the circadian clock.

## The serine loop differentially gates access to the secondary pocket of CRY1 and CRY2

To explore the structural basis for differential binding of CRY PHR domains to the CLOCK:BMAL1 PAS domain core, we examined existing crystal structures of mouse CRY1 and CRY2 PHR domains (*Michael et al., 2017a*; *Xing et al., 2013*) to identify sites of structural variability. Despite sharing 80% identity (91% homology) and having PHR domain structures with a Cα Root Mean Square Deviation (RMSD) of 1.4 Å (*Figure 3A*, *Figure 3—figure supplement 1*), molecular dynamics (MD) simulations of the PHR domains revealed that CRY1 exhibits a higher degree of flexibility at the serine loop (*Figure 3B*) and adjacent secondary pocket, which binds to CLOCK PAS-B (*Michael et al., 2017a*). In addition, another loop (E196-S207) adjacent to the serine loop showed decreased flexibility in simulations of CRY2 compared to CRY1. Proline residues at positions P219 and P225 in CRY2 replace an aspartate (D201) and serine (S207) in CRY1 at this loop, which could be responsible for the lower flexibility of this region in CRY2. The other side of the PHR domain in both CRYs also contains flexible regions, including the phosphate loop (p-loop) (*Ode et al., 2017*) and a proximal loop containing a nuclear localization signal (NLS) (*Figure 3B*, *Figure 3—figure supplement 1*).

In support of the differential dynamics observed in MD simulations, neither of the two crystal structures of the apo CRY1 PHR domain contain density for the serine loop (*Czarna et al., 2013*; *Michael et al., 2017a*). By contrast, this loop forms a short α-helix in structures of apo or FAD-bound

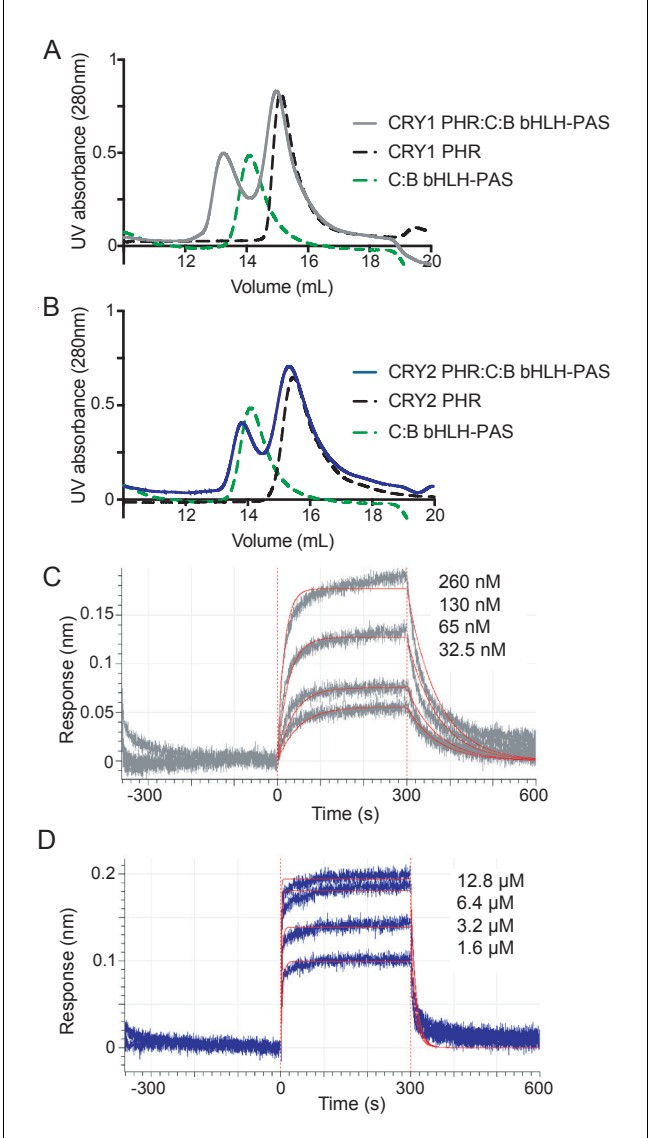

**Figure 2.** CRY1 binds more tightly to PAS domain core of CLOCK:BMAL1 than CRY2. (**A**) Gel filtration analysis of complex formation of CRY1 PHR domain mixed with the CLOCK:BMAL1 (C:B) bHLH-PAS heterodimer. CRY1 PHR domain (black), C:B bHLH-PAS (green), or CRY1 PHR domain incubated with C:B bHLH-PAS (gray) were run on a Superdex 200 10/300 GL column. B, Gel filtration analysis of complex formation of CRY2 PHR domain mixed with the CLOCK:BMAL1 (C:B) bHLH-PAS heterodimer. CRY2 PHR domain (black), C:B bHLH-PAS (green), or CRY2 PHR domain incubated with C:B bHLH-PAS (blue) were run on a Superdex 200 10/300 GL column. C-D, BLI data for CRY1 PHR domain (gray, (**C**) or CRY2 PHR domain (blue, (**D**) binding to immobilized, biotinylated CLOCK:BMAL1 PAS-AB. Inset values represent the concentrations of CRY for individual binding reactions, top to bottom. Vertical red dashed lines indicate the beginning of association and dissociation. The red solid line is the nonlinear least squares fitting. CRY1 PHR domain $K_d$ = 65 ± 6 nM; CRY2 PHR domain $K_d$ = 1.2 ± 0.2 μM (mean of two independent experiments). Data shown from one representative experiment of n = 2 assays. See *Figure 2—figure supplement 1* for additional information related to this figure.

The online version of this article includes the following figure supplement(s) for figure 2:

**Figure supplement 1.** Gel filtration analysis of CRY1, CRY2 and CLOCK:BMAL bHLH-PAS proteins.

---

CRY2 (*Xing et al., 2013*). Only two amino acids differ between CRY1 and CRY2 in this loop: G43/A61 and N46/S64 (using CRY1/CRY2 numbering, *Figure 3C*). Since the substitution of alanine for glycine at position 61 likely stabilizes the helical content of CRY2 in this loop (*López-Llano et al., 2006*), we created an in silico mutant of CRY2 swapping in the two CRY1 residues (A61G/S64N,

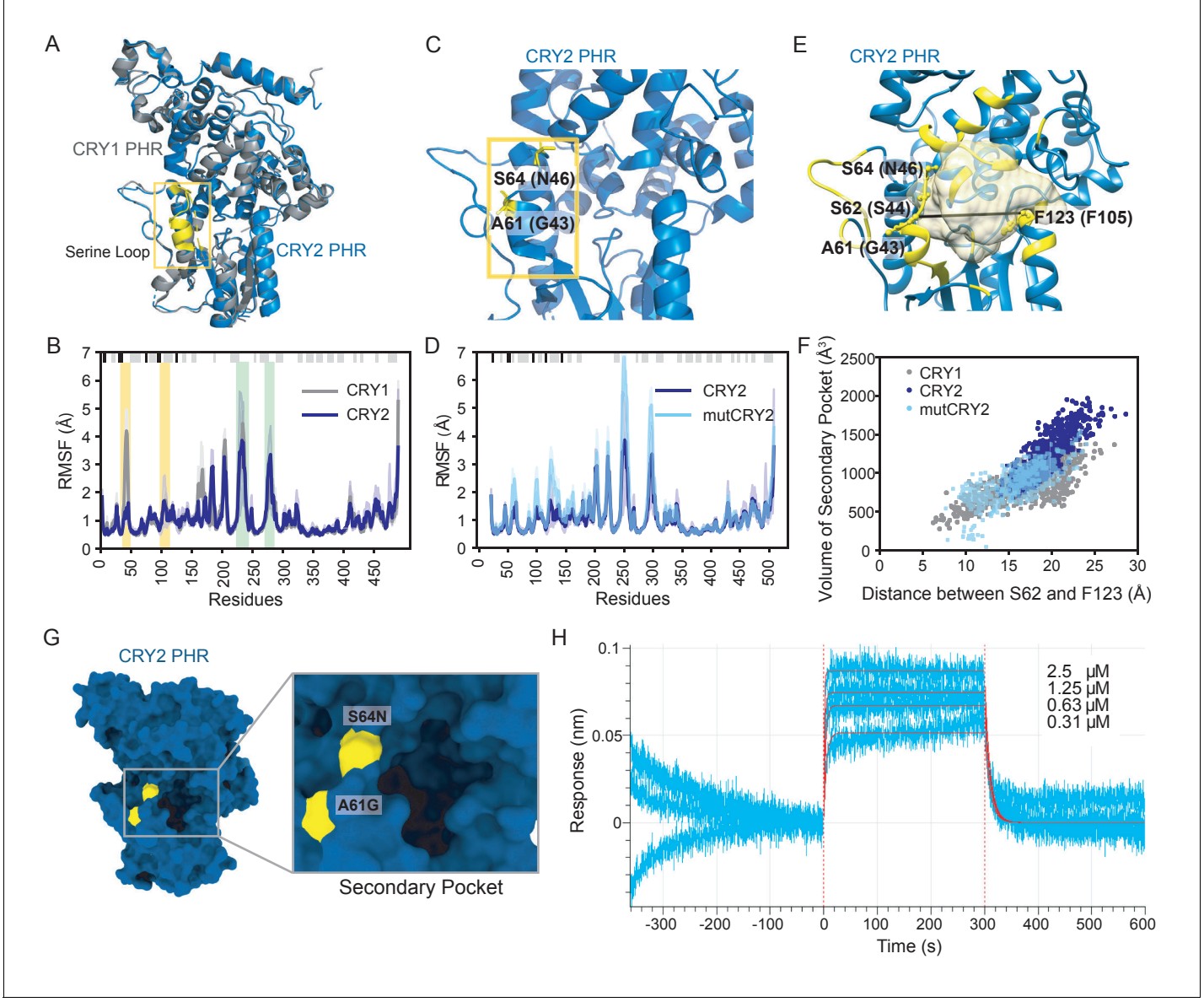

**Figure 3.** Flexibility of the serine loop controls size of the secondary pocket and CLOCK:BMAL1 binding in CRYs. (**A**) Structural alignment of CRY1 PHR domain (PDB: 5T5X) and CRY2 PHR domain (PDB: 4I6E). The serine loops are highlighted in yellow and boxed. (**B**) Root mean square fluctuation (RMSF) values obtained from MD simulations for Cα atoms of each residue in CRY1 PHR domain (gray) or CRY2 PHR domain (blue) from n = 3 independent runs. The mean RMSF is depicted in dark shades with the variation between minimum and maximum values in light shades. Secondary structure is depicted at the top of the plot (α-helix, gray; β-strand, black) and residues in the serine loop and p-loop are shaded yellow and green, respectively. (**C**) Crystal structure of CRY2 PHR domain (PDB: 4I6E) highlighting two residues that vary from CRY1 on the serine loop adjacent to the secondary pocket. (**D**) RMSF values for the wild-type CRY2 PHR domain (dark blue) or mutCRY2 (S64N/A61G, light blue), as above. (**E**) Volume of the secondary pocket (yellow cartoon and surface representation) and residues used for distance measurements of the secondary pocket opening in CRY2 (S62 and F123, depicted) or CRY1 (S44 and F105, by conservation). Black line indicates distance measured. (**F**) Scatter plot of secondary pocket volume (Å³) versus opening distance between CRY2 S62 and F123 (or S44 and F105 in CRY1). (**G**) Surface representation of the CRY2 PHR domain crystal structure (PDB: 4I6E) with the two mutations highlighted in yellow on the serine loop. (**H**) BLI data for CRY2 2M PHR domain (blue) binding to immobilized, biotinylated CLOCK:BMAL1 PAS-AB. Inset values represent the concentrations of CRY for individual binding reactions, top to bottom. Vertical red dashed lines indicate the beginning of association and dissociation. Red solid line, nonlinear least squares fitting to a one-site binding model. Calculated $K_d$ for CRY2 2M PHR domain = 343.8 ± 26.9 nM (mean ± SD from n = 2 independent assays). See *Figure 3—figure supplements 1–3* and *Supplementary file 1* for additional information related to this figure.

The online version of this article includes the following figure supplement(s) for figure 3:

**Figure supplement 1.** Kullback-Leibler divergence and Free Energy Landscape of CRY PHR domains.

**Figure supplement 2.** The CRY2 7M mutant increases affinity for the PAS domain core of CLOCK:BMAL1.

*Figure 3 continued on next page*

**Figure supplement 3.** Analysis of hydrogen bonds in CRY PHR domain serine loops.

mutCRY2) to see if this would increase flexibility of the serine loop. Comparing the Root Mean Square Fluctuations (RMSF) for wild type and mutCRY2 in MD simulations revealed that flexibility of the serine loop and secondary pocket increased in the mutant (*Figure 3D*), suggesting that minor local perturbations to the serine loop could substantially impact CRY function. Moreover, reconstitution of *Cry1*$^{-/-}$; *Cry2*$^{-/-}$; *Per2*$^{Luc}$ cells with a G43A/N46S CRY1 mutant (swapping CRY2 residues at this site into CRY1) lead to a significantly shorter period than with wild-type CRY1 (*Rosensweig et al., 2018*), consistent with a more CRY2-like, weaker repressor activity (*Hirota et al., 2012*; *Liu et al., 2007*).

To further explore the difference in the conformational ensemble of CRY1 and CRY2, we calculated the Kullback-Leibler (KL) divergence of the phi, psi, and chi1 dihedral angle distribution for all of the residues (*Figure 3—figure supplement 1*). KL divergence is used to quantitatively describe the differences in the conformational states of residues in two equilibrium ensembles (*McClendon et al., 2012*). A high KL divergence for a particular residue signifies that its conformational state shows considerable difference between the two ensembles in torsional space. Several residues throughout the CRY1 and CRY2 PHR domains show high KL divergence; in particular, the residues with the highest divergence concentrate in and around the secondary pocket, corresponding to CRY1 residues W52, Q79, D82, Y100, R127, and W390 (*Figure 3—figure supplement 1*). Notably, the comparison of conformational ensembles for wild-type CRY2 and the A61G/N64S CRY1-swapped CRY2 mutant reveals high divergence at the secondary pocket and, to a lesser extent, throughout the rest of the protein (*Figure 3—figure supplement 1*). These data suggest that the effect of substitutions in the secondary pocket that alter its rigidity may extend beyond the secondary pocket to influence the overall dynamical ensemble of CRY PHR domains.

Based on these data, alterations in the structure and dynamics of the serine loop may influence access to the secondary pocket that is located immediately adjacent to this loop. To quantify how variations in the serine loop of CRY1/CRY2 alter the conformational landscape of the secondary pocket, we calculated the volume of the pocket and the distance between center of mass of residue S62 and F123 in CRY2, located on the opposite sides of the pocket (corresponding to S44 and F105 in CRY1), for each of the independent MD trajectories (*Figure 3E*). This analysis revealed an approximately linear relationship between the interatomic distance of the loops that line the secondary pocket and its volume (*Figure 3F*). The overall distribution for CRY2 measurements clustered toward the upper right region of the plot, demonstrating that CRY2 samples a larger pocket volume (1215 ± 275 Å$^3$) as well as having larger opening to the pocket. The distribution of CRY1 measurements revealed a larger variation in distance across the pocket, likely due to the flexibility of the serine loop, but consistently had a smaller pocket volume (876 ± 217 Å$^3$, p<0.001, Wilcoxon ranksum test) compared to CRY2. As predicted from its enhanced flexibility, the in silico mutant of the CRY2 serine loop (i.e., A61G/N64S in mutCRY2) resulted in a significant change in its overall distribution towards a smaller volume of the secondary pocket (868 ± 230 Å$^3$, p<0.001, Wilcoxon ranksum test) and shorter distances across the pocket, much like we found for CRY1.

We determined the free energy landscape (FEL) of the PHR domains using the same volumes and distances as reaction coordinates to show that the low energy basin for CRY1 corresponds to the smaller volume and shorter distances as compared to the CRY2 FEL (*Figure 3—figure supplement 1*). In the case of the CRY1-like mutCRY2, the FEL shows a striking similarity to CRY1, with an expanded low energy basin and minimum corresponding to lower values of volume and distance (*Figure 3—figure supplement 1*). We further compared the volume and distances for the frames corresponding to the energy minimum in each FEL (*Supplementary file 1a*). Here again, the values corresponding to CRY1 and mutCRY2 showed a distinct similarity, whereas CRY2 had a higher value for both volume and distance. These observations suggest that decreasing the flexibility of the serine loop correlates with an increased volume of the secondary pocket, which could play a role in the decreased affinity of CRY2 for the PAS domain core of CLOCK:BMAL1.

We sought to experimentally test if mutations on the serine loop and around the secondary pocket could alter the affinity of CRY2 for the PAS domain core of CLOCK:BMAL1. We first utilized

the CRY2 7M mutant (*Rosensweig et al., 2018*), which incorporates seven mutations in and around the serine loop and the secondary pocket to swap CRY2 residues with their corresponding residues from CRY1 (*Figure 3—figure supplement 2*). We chose this mutant to characterize biochemically because genetic reconstitution of circadian rhythms in *Cry1⁻/⁻; Cry2⁻/⁻; Per2^Luc* cells demonstrated that CRY2 7M was sufficient to largely recapitulate a CRY1-like repression phenotype and period (*Rosensweig et al., 2018*). In line with this cell-based observation, we found that the CRY2 7M mutant had a 10-fold enhancement in affinity for the CLOCK:BMAL PAS domain core, as determined by BLI (*Figure 3—figure supplement 2*). To further establish the contribution of substitutions on serine loop to CLOCK:BMAL1 binding, we purified the CRY2 2M mutant corresponding to mutCRY2 (A61G/N64S) and measured its affinity for the CLOCK:BMAL1 PAS domain core. As predicted by our MD simulations, we found that simply swapping two residues on the serine loop increased CRY affinity for CLOCK:BMAL1 by ~4 fold (*Figure 3G–H*).

To characterize interactions at the serine loop that might be responsible for its conformational diversity, we measured the occurrence of hydrogen bonds formed by loop residues during the MD simulations. A comparison of frequently occurring hydrogen bonds among apo CRY1, CRY2 and mutCRY2 revealed that the serine loop in CRY2 is stabilized primarily by hydrogen bonds *within* the loop residues, as might be expected from the short helical structure present in the starting model. On the other hand, we observed fewer hydrogen bonds within the loop in CRY1 and mutCRY2 trajectories (*Supplementary file 1b*). Furthermore, we observed that N64 forms hydrogen bonds with D400 and E401 in mutCRY2 that are not frequently seen in the CRY2 simulations. Interestingly, in its native environment, N46 of CRY1 also forms hydrogen bonds with its corresponding residues at these positions (E382 and E383) with high frequency (*Figure 3—figure supplement 3*). Collectively, these results establish that the A61G/N64S substitutions make the CRY2 serine loop behave more like CRY1 and significantly alter its ability to bind CLOCK:BMAL1, helping to differentiate the roles of CRY1 and CRY2 as circadian repressors in the mammalian circadian clock.

## The CRY1 serine loop is disordered in the CRY1 PHR:PER2 CBD complex

In addition to binding near the BMAL1 TAD-binding site at the CC helix of CRY1 and CRY2 (*Figure 1D*), the PER2 CBD also wraps around the distal side of the PHR domain to come into close contact with the serine loop (*Nangle et al., 2014*; *Schmalen et al., 2014*; *Figure 4A*). Because of its proximity to the secondary pocket, we wanted to see how PER2 might regulate the affinity of CRY1 or CRY2 for CLOCK:BMAL1. However, the previous crystal structure of a CRY1 PHR:PER2 CBD complex (PDB: 4CT0) revealed that a portion of the PER2 peptide encoded by the expression vector packed against the secondary pocket, resulting in substantial ordering of the CRY1 serine loop in the secondary pocket (*Schmalen et al., 2014*; *Figure 4—figure supplement 1*). To see if binding of the PER2 CBD truly alters the serine loop and secondary pocket of CRY1, we determined a crystal structure of the CRY1 PHR:PER2 CBD complex that eliminated this artifact (*Figure 4*, *Figure 4—figure supplement 1*, *Supplementary file 2a*). The overall conformation of the PER2 CBD in our structure is highly similar to the previously solved structure (0.76 Å Cα RMSD over PER2 residues 1136–1207). However, we now see that the serine loop maintains most of its flexibility in the PER2-bound state.

To gain further insight into the structural organization of the PER2 CBD:CRY1 PHR:CLOCK PAS-B complex, we generated HADDOCK models of the CRY1 PHR:CLOCK PAS-B domain complex using representative CRY1 structures from the MD simulations (with the serine loop intact) and then superimposed the structures from the two largest and best scoring clusters from HADDOCK onto our CRY1 PHR:PER2 CBD crystal structure (*Figure 4—figure supplement 2*, *Supplementary file 2b*). While both models clearly place the HI loop of the CLOCK PAS-B domain into the secondary pocket of CRY1, they differ in the angular orientation of CLOCK PAS-B with respect to CRY1 by ~30°. We previously observed two similar models of the CRY1 PHR: CLOCK PAS-B complex by HADDOCK, but couldn't resolve the ambiguity between them using low resolution small-angle x-ray scattering data (*Michael et al., 2017a*). To address this here, we compared simulated electron microscopy projections of these two clusters to reference-free 2D class averages obtained from imaging of a PER2 CBD:CRY1 PHR:CLOCK:BMAL1 complex by cryo-EM (*Figure 4—figure supplement 2*). While the CLOCK PAS-B domain was more flexible relative to the CRY1 PHR domain, the overall position of CLOCK PAS-B in Cluster 1 agrees much better with the experimental data than Cluster 2. Therefore,

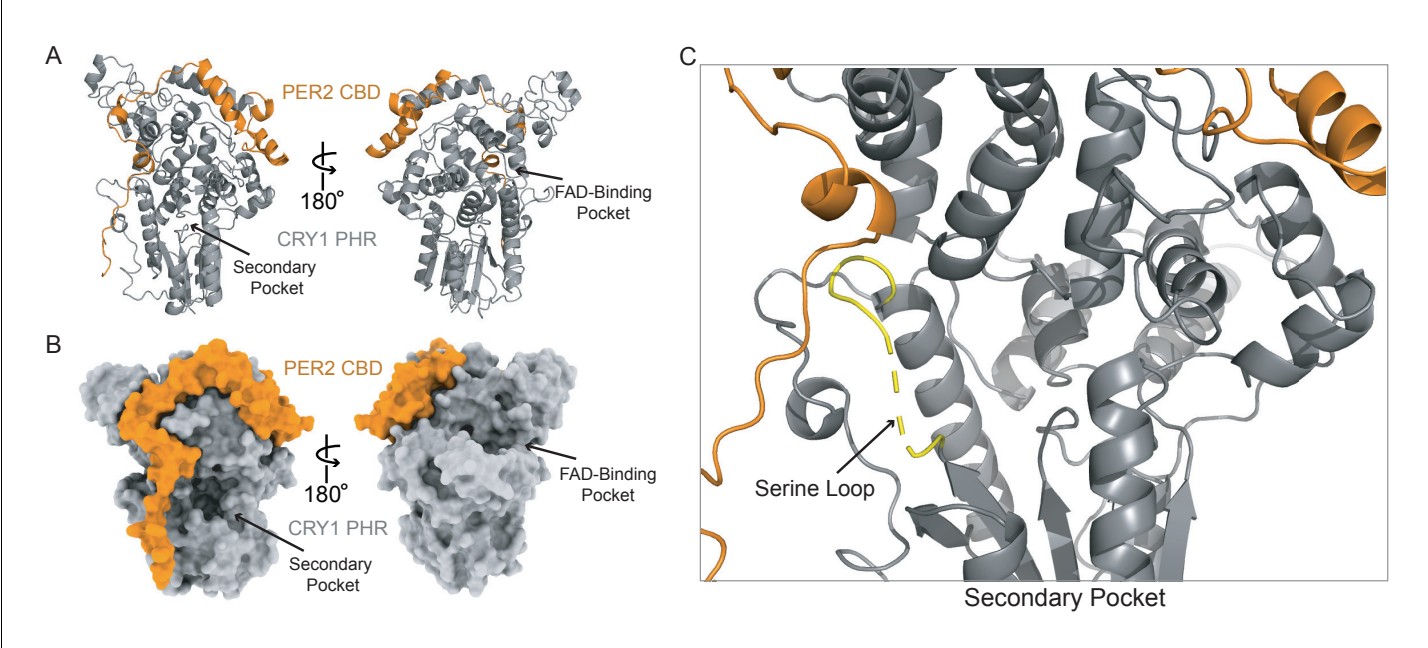

**Figure 4.** A new CRY1 PHR:PER2 CBD crystal structure. (A-B) Crystal structure of the CRY1 PHR:PER2 CBD complex (PDB: 6OF7) with CRY1 PHR domain (gray) and PER2 CBD (orange). Ribbon and surface representations on the left highlight the secondary pocket, while the FAD binding pocket is shown on the right. (B) Zoomed in view of the secondary pocket showing the partially ordered serine loop (yellow). See *Figure 4—figure supplements 1–3* and *Supplementary file 2* for additional information related to this figure.

The online version of this article includes the following figure supplement(s) for figure 4:

**Figure supplement 1.** Additional views of the new CRY1 PHR:PER2 CBD crystal structure.

**Figure supplement 2.** Representative HADDOCK models of the PER2 CBD:CRY1 PHR:CLOCK PAS-B complex and comparison to 2D class averages from cryo-EM data.

**Figure supplement 3.** CRY PHR binding to PAS domain core with CLOCK W362A mutation.

the integration of computational docking and experimental data support a model of PER2 CBD: CRY1 PHR:CLOCK PAS-B complex.

We previously showed that mutation of W362 to alanine in the HI loop of CLOCK PAS-B reduced binding of the CRY1 PHR and significantly decreased CRY1 repression of CLOCK:BMAL1 in 293 T cells (*Michael et al., 2017a*). To examine how this mutation quantitatively influences binding of CLOCK to CRY1 and CRY2, we performed BLI-based binding studies using the CLOCK:BMAL1 PAS domain core harboring the W362A CLOCK mutant (*Figure 4—figure supplement 3*). The PHR domains of CRY1 and CRY2 both demonstrated a significantly decreased affinity for the mutant PAS domain core, with a $K_d$ for CRY1 of 6.6 ± 2.6 μM, reduced nearly 100-fold from the wild-type PAS domain core, while CRY2 bound with ~10 fold reduced affinity ($K_d$ = 10.2 ± 0.2 μM). These results confirm the requirement for CLOCK PAS-B in interaction of the CLOCK:BMAL1 PAS domain core with both the CRY1 and CRY2 PHR domains, specifically highlighting the critical role of the tryptophan residue on the HI loop of the CLOCK PAS-B domain.

### The PER2 CBD remodels the CRY2 serine loop to promote binding to the PAS domain core of CLOCK:BMAL1

A recent study showed that association of CRY2 with PER2 helps it to form a stable complex with CLOCK:BMAL1, while CRY1 can bind stably to CLOCK:BMAL1 either with or without the PER proteins (*Rosensweig et al., 2018*), laying the foundation to understand the unique role of CRY1 in the late repressive complex (*Koike et al., 2012*; *Partch et al., 2014*). To explore the structural basis for this model, we asked if PER2 binding could regulate affinity of the CRY1 and CRY2 PHR domain for the PAS domain core of CLOCK:BMAL1. Comparing the secondary binding pocket of the apo CRY1 PHR domain structure to our CRY1 PHR:PER2 CBD structure (*Figure 5A*), we observed that PER2 CBD induces a modest structuring at the C-terminal end of the loop (residues 44–47). This

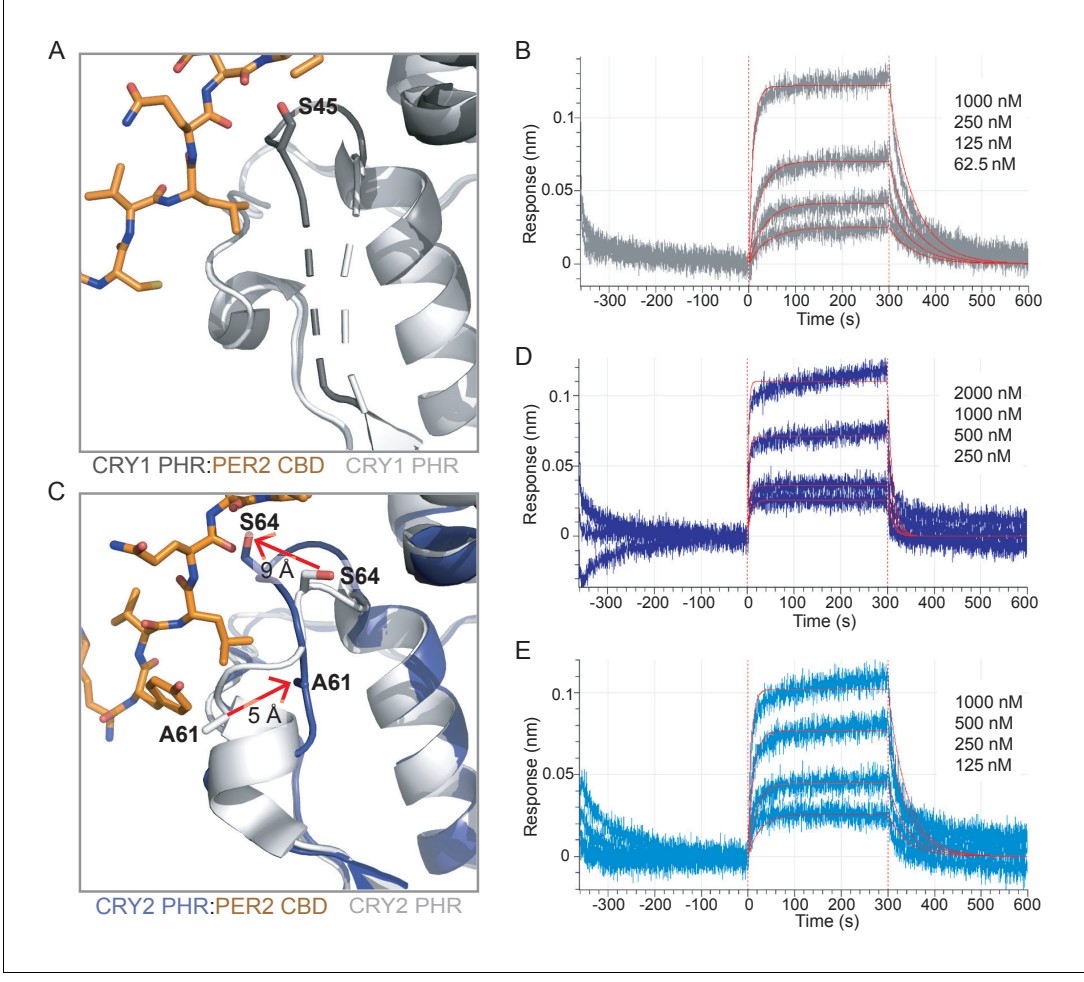

**Figure 5.** The PER2 CBD alters the CRY serine loops to modulate their affinity for the CLOCK:BMAL1 PAS core. (**A**) Comparison of the serine loop from apo CRY1 PHR domain (light gray, PDB: 5T5X) and the CRY1 PHR:PER2 CBD complex (dark gray:orange, PDB: 6OF7). A slight structuring of the C-terminus of the serine loop occurs upon addition of PER2 CBD, while the N-terminus remains flexible (dashed line). (**B**) BLI data for the CRY1 PHR:PER2 CBD complex (gray) binding to immobilized, biotinylated CLOCK:BMAL1 PAS-AB. Inset values represent the concentrations of CRY for individual binding reactions, top to bottom. Vertical red dashed lines indicate the beginning of association and dissociation. Red solid line, nonlinear least squares fitting to a one-site binding model. Calculated $K_d$ for CRY1 PHR:PER2 CBD = 196 ± 34 nM (mean ± SD from n = 2 independent assays). (**C**) Comparison of the serine loop from the apo CRY2 PHR domain (light gray, PDB: 4I6E) and the CRY2 PHR:PER2 CBD complex (blue:orange, PDB: 4U8H). (**D**) BLI data for the CRY2 PHR:PER2 CBD complex (dark blue) binding to immobilized, biotinylated CLOCK:BMAL1 PAS-AB domains. Calculated $K_d$ for CRY2 PHR:PER2 CBD = 604 ± 29 nM (mean ± SD from n = 2 independent assays). (**E**) BLI data for the CRY2 7M PHR:PER2 CBD complex (light blue) binding to immobilized, biotinylated CLOCK:BMAL1 PAS-AB. Calculated $K_d$ for CRY2 7M PHR:PER2 CBD = 159 ± 66 nM (mean ± SD from n = 2 independent assays). See *Figure 5—figure supplements 1–3* for additional information related to this figure.

The online version of this article includes the following figure supplement(s) for figure 5:

**Figure supplement 1.** Binding to PER2 CBD rearranges the serine loops on CRY1 and CRY2.

**Figure supplement 2.** Formation of stable CRY PHR domain:PER2 CBD complexes for binding studies.

**Figure supplement 3.** The PER2 CBD does not bind directly to the PAS domain core of CLOCK:BMAL1.

reorganization is stabilized by an interaction between S45 on CRY1 and the backbone of Q1135 on the PER2 CBD (*Figure 5A*, *Figure 5—figure supplement 1*). To determine if addition of the PER2 CBD alters binding between the CRY1 PHR domain and CLOCK:BMAL1, we utilized BLI binding assays using preformed CRY:PER2 CBD complexes with the immobilized PAS domain core and

found that the CRY1:PER2 CBD complex binds with a ~ 3 fold reduction in affinity (*Figure 5B*, *Figure 5—figure supplement 2*). Therefore, PER2 modestly weakens interaction of the CRY1 PHR domain with the CLOCK:BMAL1 PAS core, suggesting that this modest ordering of the CRY1 serine loop makes docking of the CLOCK PAS-B into the secondary pocket less favorable.

By contrast, our comparison of the apo CRY2 PHR domain structure with the CRY2 PHR:PER2 CBD complex revealed a major structural reorganization of the serine loop induced by PER2 (*Figure 5C*). As described above, the beginning of the serine loop in the apo CRY2 PHR domain is structured with a short α-helix and a stable but extended loop at the end (*Nangle et al., 2014*; *Xing et al., 2013*). However, in the CRY2 PHR: PER2 CBD complex, the end of the serine loop is stabilized in an alternate position by multiple hydrogen bonds between S64 on CRY2 and A1135 and D1136 of the PER2 CBD (*Figure 5—figure supplement 1*). This interaction moves S64 of CRY2 out of the secondary pocket by ~9 Å (*Figure 5C*), largely mimicking the PER2-dependent interaction and subsequent loop orientation we observed with S45 on CRY1 (*Figure 5—figure supplement 1*). Additionally, the PER2 CBD remodels the N-terminus of the serine loop, causing an unfolding of the short α-helix and a translation of A61 by ~5 Å towards the secondary pocket (*Figure 5C*). Therefore, the addition of PER2 CBD causes the serine loops of CRY1 and CRY2 to take on a largely similar conformation, with the loop in CRY1 becoming more structured and the loop in CRY2 becoming less structured.

Based on the rearrangement of the CRY2 serine loop induced by PER2, we anticipated that PER2 would enhance affinity of CRY2 for the PAS domain core of CLOCK:BMAL1.

We used BLI binding assays to assess binding of a CRY2:PER2 CBD complex to the immobilized PAS domain core of CLOCK:BMAL1 and observed an increase in affinity of CRY2 for CLOCK:BMAL1 by 2-fold (*Figure 5D*). While this is a modest increase in affinity, these data are consistent with the observation that addition of PER2 enhances co-immunoprecipitation of CLOCK:BMAL1 by CRY2 (*Rosensweig et al., 2018*). We also probed for a direct interaction of the PER2 CBD with the PAS domain core of CLOCK:BMAL1 by BLI, but observed no detectable binding (*Figure 5—figure supplement 3*), demonstrating that the effect of PER2 CBD is likely due to regulation of the CRY serine loop and not a via direct interaction with CLOCK:BMAL1. These results provide a mechanistic foundation for the model where formation of a complex with PER2 makes CRY2 a stronger repressor by enhancing its affinity for CLOCK:BMAL1 (*Rosensweig et al., 2018*).

Based on the structural and computational analyses presented thus far, we attribute this tighter binding to CLOCK:BMAL1 to the loss of helical structure in the serine loop observed when CRY2 is bound to the PER2 CBD. Therefore, the CRY2 7M mutant, which substitutes CRY1 residues into the serine loop and secondary pocket, might not be expected to exhibit a gain in affinity for CLOCK: BMAL1 in the presence of the PER2 CBD. To test this prediction, we collected BLI binding data on the CRY2 7M PHR domain mutant in the presence of PER2 CBD. In line with our model, the CRY2 7M:PER2 CBD complex behaved more like CRY1 and exhibited a similar drop in affinity for CLOCK: BMAL1 (*Figure 5E*). Altogether, our study establishes that the PER2 CBD tunes affinity of the CRY PHR domains for the PAS domain core of CLOCK:BMAL1 through differential interactions at the serine loop. Notably, when CRY1 or CRY2 are associated with the co-repressor PER2, their affinity for CLOCK:BMAL1 is at its most similar (within a ~ 3 fold difference) compared to the 20-fold difference that we observe with the two proteins in the absence of the PER2 CBD.

## Discussion

Since the identification of core mammalian clock proteins nearly two decades ago, numerous studies have revealed how they work at the network level to generate the transcriptional feedback loop that confers circadian timekeeping (*Takahashi, 2017*). However, we still lack a mechanistic understanding of how most clock proteins interact with and/or regulate one another to fulfill their important roles as dedicated molecular 'cogs' in the circadian clock. In this study, we integrated biochemistry, structural biology, and computational methods to determine how the association of CLOCK:BMAL1 with CRY1 or CRY2 is differentially regulated by dynamics at the serine loop on their respective PHR domains. In contrast to CRY2, CRY1 has a flexible serine loop adjacent to its secondary pocket on the PHR domain that confers tight binding to the PAS domain core of CLOCK:BMAL1. Substitutions to CRY1 residues at serine loop and/or secondary pocket in CRY2 are sufficient to enhance its association with the PAS domain core of CLOCK:BMAL1, in line with a previous study showing that these

mutants largely recapitulated isoform-dependent changes in circadian period (*Rosensweig et al., 2018*). Remarkably, we found that binding of the PER2 CBD remodels the ordered serine loop of CRY2 to enhance its affinity for CLOCK:BMAL1, conferring CRY1-like affinity and dynamics to the serine loop. Therefore, small changes in the structure and dynamics of the serine loop with and without PER2 allow cryptochromes to fine-tune their repressive power by controlling how tightly they bind to CLOCK:BMAL1.

Our data suggest that the differential effect of the PER2 CBD on CRY PHR binding to CLOCK:BMAL1 allows it to serve as a molecular equalizer—PER2 compensates for poorer binding by CRY2 by modestly enhancing its affinity for the transcription factor, while at the same time reducing CRY1 binding to bring the overall affinities of the two cryptochromes for the PAS domain core of CLOCK:BMAL1 to similar levels. This suggests that the actions of CRY1 and CRY2 within the heteromultimeric PER-CRY repressive complexes (*Aryal et al., 2017*; *Lee et al., 2001*) may be largely similar. The apparent dependency of CRY2 on PER1/2, the PER isoforms that possess a functional CRY-binding domain (*Miyazaki et al., 2001*; *Yagita et al., 2002*), is consistent with the expression of CRY2 that occurs in phase with the PER proteins. By contrast, the peak expression of CRY1 is delayed by several hours from the other repressors (*Lee et al., 2001*; *Ukai-Tadenuma et al., 2011*). The same temporal profiles are observed in the genome-wide mapping of their DNA occupancy with CLOCK:BMAL1, revealing two distinct waves of repressive complexes: 'early' complexes that contain PER1/2 proteins, their associated kinase CK1δ, CRY2, and some CRY1 with a variety of non-stoichiometrically associated epigenetic regulators (*Aryal et al., 2017*; *Kim et al., 2015*), and a 'late' complex consisting of CRY1 bound to CLOCK:BMAL1 in the apparent absence of PER proteins (*Koike et al., 2012*; *Partch et al., 2014*). The 20-fold increase in affinity of CRY1 that we observed for the PAS domain core of CLOCK:BMAL1 relative to CRY2 therefore likely plays a critical role in its ability to participate in the PER-free 'late' complex, and may also give rise to the CRY1-selective extension of the repression phase observed in the *Fbxl3^Afh* mutant (*Anand et al., 2013*). Altogether, these data suggest that a proper balance of the biochemically distinct PER-containing early and PER-free late repressive complexes work together to generate the circadian period of ~24 hr.

Understanding precisely why the more dynamic serine loop helps CRY1 bind to the PAS domain core of CLOCK:BMAL1 awaits determination of a higher resolution structure of the complex. Like many other transcription factors, CLOCK:BMAL1 is a 'malleable machine' (*Fuxreiter et al., 2008*) and its dynamic nature presents challenges for structure determination, particularly its flexibly tethered, modular PAS domains and disordered regions that play important functional roles (*Michael et al., 2017a*; *Xu et al., 2015*). From the perspective of cryptochromes, the dynamic nature of the serine loop may be associated with allosteric changes throughout the PHR domain and evolution of transcriptional repressor function. Notably, the secondary pocket in photoreceptive cryptochromes from *Arabidopsis* and *Drosophila* (*Brautigam et al., 2004*; *Levy et al., 2013*), as well as the evolutionarily-related DNA repair enzyme photolyase (*Hitomi et al., 2009*; *Park et al., 1995*) is considerably smaller in volume, where it binds to small molecule cofactors that facilitate light harvesting (*Sancar, 2003*). The ability of repressor-type cryptochromes to interact with CLOCK PAS-B appears to be deeply rooted in evolution of the metazoan circadian clock, because vertebrate-like cryptochromes that act as transcriptional repressors in insects (*Zhu et al., 2005*) like the monarch butterfly also depend to the same extent on multivalent interactions with CLOCK PAS-B and the BMAL1 TAD as they do in mammals (*Sato et al., 2006*; *Xu et al., 2015*; *Zhang et al., 2017*).

The BMAL1 TAD is the primary driver of transcriptional activation by CLOCK:BMAL1 in mammalian and insect systems with repressor-type cryptochromes, as truncation of the TAD decimates CLOCK:BMAL1 activity and leads to arrhythmicity in vivo (*Kiyohara et al., 2006*; *Park et al., 2015*; *Zhang et al., 2017*). Sequestration of the TAD by repressor-type cryptochromes allows them to compete with coactivators at a highly conserved and overlapping binding motif in the TAD (*Gustafson et al., 2017*; *Xu et al., 2015*). Strikingly, we did not observe differential binding to the BMAL1 TAD by CRY1 or CRY2, either in the absence or presence of the PER2 CBD. However, it remains to be seen if other regions in the disordered BMAL1 C-terminus (i.e., the G region *Xu et al., 2015*) or CRY C-termini (*Czarna et al., 2011*; *Partch et al., 2005*) also participate in CRY binding. This suggests that the changes in CRY affinity for the PAS domain core of CLOCK:BMAL1 that we observed here underlie the major functional differences between CRY1 and CRY2 that contribute to differential period regulation (*Edwards et al., 2016*; *van der Horst et al., 1999*; *Vitaterna et al.,*

1999) and the ability of CRY1 to sustain cellular circadian rhythms (*Liu et al., 2007*). Therefore, cryptochrome recruitment to the PAS domain core likely represents a key regulatory node in the clock.

## Materials and methods

### Protein expression and purification

Using the baculovirus expression system (Invitrogen), the following constructs were expressed in Sf9 suspension insect cells (Expression Systems): His-tagged mouse CRY1 PHR domain (residues 1–491), CRY2 PHR domain (residues 1–512), CRY2 7M PHR domain (residues 1–512 with the following mutations: A61G, S64N, S394E, V396M, R397K, D400E, F408W), CRY2 2M PHR domain (residues 1–512 with the following mutations: A61G, S64N) CLOCK:BMAL1 bHLH-PAS domains (CLOCK residues 26–395, BMAL1 residues 62–441) or GST-tagged BMAL1 PAS-AB domains (residues 136–441). Sf9 suspension cells were infected with a P3 virus at $1.2 \times 10^6$ cells per milliliter and grown for 72 hr at 27°C with gentle shaking. Cells were centrifuged at 4°C for 4000 x rpm for 15 min. CRY1, CRY2, CRY2 2M and CRY2 7 M cells were resuspended in 50 mM Tris buffer pH 7.5, 300 mM NaCl, 20 mM imidazole, 10% (vol/vol) glycerol, 0.1% (vol/vol) Triton X-100, 5 mM β-mercaptoethanol and EDTA-free protease inhibitors (Pierce). Cells were lysed using a microfluidizer followed by sonication with a ¼" probe on ice for 15 s on, 45 s off for three pulses at 40% amplitude. Lysate was clarified at 4°C for 19,000 rpm for 45 min. The protein was isolated by $Ni^{2+}$-nitrilotriacetic acid (Ni-NTA) affinity chromatography (Qiagen). Proteins were eluted with 50 mM Tris buffer pH 7.5, 300 mM NaCl, 250 mM imidazole, 10% (vol/vol) glycerol, and 5 mM β-mercaptoethanol. Proteins were further purified by HiTrap SP cation exchange chromatography (GE Healthcare) and Superdex 75 gel filtration chromatography (GE Healthcare) into 20 mM HEPES buffer pH 7.5, 125 mM NaCl, 5% (vol/vol) glycerol, and 2 mM tris(2-carboxyethyl)phosphine (TCEP).

CLOCK:BMAL1 bHLH-PAS expressing cells were resuspended in 20 mM sodium phosphate buffer pH 8, 15 mM imidazole, 10% (vol/vol) glycerol, 0.1% (vol/vol) Triton X-100 and 5 mM β-mercaptoethanol. Cells were lysed, clarified and Ni-NTA affinity purification was performed as described above. The complex was eluted with sodium phosphate buffer pH 8, 250 mM imidazole, 10% (vol/vol) glycerol and 5 mM β-mercaptoethanol. The complex was further purified by HiTrap Heparin HP affinity column (GE Healthcare) after diluting eluent ~5 fold with 20 mM sodium phosphate buffer pH 7.5, 50 mM NaCl, 2 mM Dithiothreitol, and 10% (vol/vol) glycerol and loaded onto the column. After washing with five column volumes of the above buffer, the complex was then eluted with a gradient 0–100% of 20 mM sodium phosphate buffer pH 7.5, 2 M NaCl, 2 mM Dithiothreitol, and 10% (vol/vol) glycerol. The complex was further purified by Superdex 200 gel filtration chromatography (GE Healthcare) into 20 mM HEPES buffer pH 7.5, 125 mM NaCl, 5% (vol/vol) glycerol, and 2 mM TCEP.

GST-BMAL1 PAS-AB-expressing Sf9 cells were resuspended in 50 mM HEPES buffer pH 7.5, 300 mM NaCl, 5% (vol/vol) glycerol and 5 mM β-mercaptoethanol. Cells were lysed and clarified as described above. The soluble lysate was bound in batch-mode to Glutathione Sepharose 4B (GE Healthcare), then washed and eluted with 50 mM HEPES buffer pH 7.5, 150 mM NaCl, 5% (vol/vol) glycerol and 5 mM β-mercaptoethanol, 25 mM reduced glutathione. The protein was desalted into HEPES buffer pH 7, 150 mM NaCl, 5% (vol/vol) glycerol and 5 mM β-mercaptoethanol using a HiTrap Desalting column (GE Healthcare) and the GST tag was cleaved with GST-TEV protease overnight at 4°C. The cleaved GST-tag and GST-tagged TEV protease was removed by Glutathione Sepharose 4B (GE Healthcare) and the remaining BMAL1 PAS-AB protein was further purified by Superdex 75 gel filtration chromatography (GE Healthcare) into 20 mM HEPES buffer pH 7.5, 125 mM NaCl, 5% (vol/vol) glycerol, and TCEP.

Recombinant baculoviruses expressing GST-CRY2 PHR domain, His-CLOCK bHLH-PAS, and His-BMAL1 full-length were co-infected into monolayer High Five insect cells (Invitrogen) for protein expression. Cells were harvested 48 hr after infection, and lysed in a buffer containing 40 mM HEPES pH 7.5, 300 mM NaCl, 10% (v/v) glycerol, 5 mM β-mercaptoethanol. Lysed cells were clarified by ultracentrifugation, and the soluble lysate was purified over a glutathione affinity column (GE Healthcare). Bound complex was eluted via overnight on-column TEV cleavage to remove the GST- and His-tags. Eluted material was further purified by a HiTrap Q-HP (GE Healthcare) anion exchange column, followed by a Superdex 200 gel filtration column (GE Healthcare) equilibrated in 20 mM HEPES pH 7.5, 300 mM NaCl, 10% (v/v) glycerol, 1 mM DTT.

The CLOCK PAS-AB (residues 93–395) and PER2 CBD (residues 1095–1215) were expressed in Rosetta2 (DE3) *E. coli*. Protein expression was induced with 0.5 mM isopropyl-β-D-thiogalactopyranoside (IPTG) at an $OD_{600}$ of ~0.8, after which cells were grown for an additional 18 hr at 18°C. Cells were harvested by centrifugation at 4°C for 4000 x rpm for 15 min. For CLOCK PAS-AB (wild-type or W362A mutant), the protein was expressed as a fusion protein with the solubilizing tag His-NusA-XL and an N-terminal biotin acceptor peptide (BAP) (Beckett et al., 1999) C-terminal to the TEV site. Cells were resuspended in 50 mM Tris buffer pH 7.5, 300 mM NaCl, 20 mM imidazole, 10% (vol/vol) glycerol and 5 mM β-mercaptoethanol and lysed by microfluidizer. Lysate was clarified at 4°C for 19,000 rpm for 45 min. The protein was isolated by Ni-NTA affinity chromatography (Qiagen) and eluted with 50 mM Tris buffer pH 7.5, 300 mM NaCl, 250 mM imidazole, 10% (vol/vol) glycerol, and 5 mM β-mercaptoethanol. The protein was then desalted into 50 mM Tris buffer pH 7.5, 150 mM NaCl, 5% (vol/vol) glycerol and 5 mM β-mercaptoethanol, and the His-NusA-XL tag was cleaved with His-TEV protease overnight at 4°C. The cleaved tag and protease were removed by Ni-NTA affinity chromatography (Qiagen) and CLOCK PAS-AB was further purified by Superdex 75 gel filtration chromatography (GE Healthcare) into 20 mM HEPES buffer pH 7.5, 125 mM NaCl, 5% (vol/vol) glycerol, and 2 mM TCEP. PER2 CBD was expressed as a fusion protein with GST and purified as described above. All proteins were flash frozen in small aliquots and stored at −70 °C.

## Biotinylation of CLOCK PAS-AB

100 µM BAP-CLOCK PAS-AB (wild-type or W362A mutant) in 20 mM HEPES buffer pH 7.5, 125 mM NaCl, 5% (vol/vol) glycerol, and 2 mM TCEP was incubated at 4 °C overnight with 2 mM ATP, 1 µM GST-BirA (purified from *E. coli* according to a prior protocol Fairhead and Howarth, 2015) and 150 µM biotin. GST-BirA was removed afterwards with Glutathione Sepharose 4B (GE Healthcare) resin and excess biotin was separated from the labeled protein by Superdex 75 gel filtration chromatography (GE Healthcare) in 20 mM HEPES buffer pH 7.5, 125 mM NaCl, 5% (vol/vol) glycerol, and 2 mM TCEP.

## Assembling and verifying protein complexes for binding studies

To assemble CRY PHR:PER2 CBD complexes used for binding studies with CLOCK:BMAL1, a slight molar excess of zinc chloride was added to a 1:1 mix of the PER2 and CRY proteins (Nangle et al., 2014; Schmalen et al., 2014). The assembly and purity of complexes from peak fractions of the Superdex 200 10/300 GL gel filtration column (GE Healthcare) was assessed by SDS-PAGE gel electrophoresis and SimplyBlue SafeStain (Invitrogen) staining before using them in binding assays.

## Fluorescence polarization

A peptide of the minimal BMAL1 TAD (residues 594–626) was purchased from Bio-Synthesis with a 5,6-TAMRA fluorescent probe covalently attached to the N terminus. Binding assays with CRY1 PHR domain, CRY1 PHR:PER2 CBD, CRY2 PHR domain and CRY2 PHR:PER2 CBD were performed in 50 mM Bis-Tris Propane buffer pH 7.5, 100 mM NaCl, 2 mM TCEP and 0.05% (vol/vol) Tween-20. The BMAL TAD probe was diluted to a working concentration of 50 nM in assay buffer and binding was monitored by changes in fluorescence polarization with an EnVision 2103 multilabel plate reader (Perkin Elmer) with excitation at 531 nm and emission at 595 nm. Equilibrium binding dissociation constants ($K_d$) were calculated by fitting the dose-dependent change in millipolarization (mP) to a one-site specific total binding model in Prism 7 (GraphPad). The mP values shown represent the average of duplicate samples from a representative experiment of n = 3 independent assays. The $K_d$ reported (± SD) is the average of determined from the three independent assays.

## Bio-layer interferometry

BLI experiments were performed using an 8-channel Octect-RED96e (ForteBio) with a BLI assay buffer of 20 mM HEPES buffer pH 7.5, 125 mM NaCl, 5% (vol/vol) glycerol and 2 mM TCEP. All experiments began with a reference measurement to establish a baseline in BLI buffer for 120 s. Next, 1.5 µg/mL biotinylated CLOCK:BMAL1 PAS-AB (wild-type or W362A mutant in BLI buffer) was loaded on a streptavidin tip for 300 s at room temperature. Subsequently, a 360 s blocking step was performed with 0.5 mg/mL BSA, 0.02% (vol/vol) Tween, 20 mM HEPES buffer pH 7.5, 125 mM NaCl, 5% (vol/vol) glycerol and 2 mM TCEP. Association was then measured for 300 s for eight different

concentrations of the analyte (CRY, CRY:PER2 CBD, PER2 CBD) in a serial dilution starting at approximately 10x the estimated $K_d$ in blocking buffer. Dissociation was measured for 300 s in blocking buffer. Each experiment was repeated with tips that were not loaded with CLOCK:BMAL PAS-AB to provide a reference for non-specific binding to the tip. Data were processed and fit using Octet software v.7 (ForteBio). Before fitting, all datasets were reference-subtracted, aligned on the y-axis and aligned for interstep correction through their respective dissociation steps according to the manufacturer's instructions. For each experiment, at least four different concentrations were used to fit association and dissociation globally using a 1:1 binding model in Octet software v.7 (ForteBio). Ultimately, the goodness of fit was determined using $\chi^2$ and $R^2$ values according to the manufacturer's instructions.

## Molecular dynamics simulations

The crystal structure of apo CRY1 PHR domain (PDB: 5T5X) and apo CRY2 PHR domain (PDB: 4I6E) were prepared for starting models for the molecular dynamics (MD) simulations by modeling nonterminal missing residues using the Prime program (Schrödinger). All crystallographically-defined water molecules were removed from the structures. In order to prepare the system with the in silico mutated CRY2 PHR domain (A61G/S64N CRY2), the corresponding amino acids were mutated in wild type apo CRY2 PHR domain using UCSF-Chimera (*Pettersen et al., 2004*) and used as starting models. Amber99sb-ildn force field was used for simulation (*Lindorff-Larsen et al., 2010*). The structures were solvated in a dodecahedron box using TIP3P water molecules. The systems were neutralized and then $Na^+$ and $Cl^-$ ions were added to maintain a physiological ionic concentration of 0.15 M. Energy minimization was performed for the systems until the maximum force on any atom was less than 1,000 kJ $mol^{-1}nm^{-1}$ in the case of apo CRY1 and CRY2 PHR domains, and 500 kJ $mol^{-1}nm^{-1}$ in the case of the in silico mutated CRY2 PHR domain.

After energy minimization, the systems were equilibrated in an NVT ensemble for 500 ps. The temperature was maintained at 310 K using the modified Berendsen Thermostat (V-rescale) (*Bussi et al., 2007*). After this, the systems were equilibrated in an NPT ensemble for 500 ps. The pressure was maintained at 1 bar using a Berendsen barostat (*Berendsen et al., 1984*). This was followed by a production run of 500 ns for each system in an NPT ensemble using a Parinello-Rahman barostat (*Parrinello and Rahman, 1981*) and modified Berendsen Thermostat (V-rescale) with the temperature maintained at 310 K and pressure at 1 bar. Three independent simulations were completed for the apo CRY1 PHR domain and apo CRY2 PHR domain systems described in the previous section and two independent simulations were run for the in silico mutated CRY2 PHR domain system. All MD simulations were performed using Gromacs 5 (*Abraham et al., 2015*).

The trajectories were analyzed using in built functions of Gromacs. RMSD, RMSF, distances and dihedral angles were calculated using gmx rmsd, gmx rmsf, gmx distance and gmx chi programs of Gromacs respectively. UCSF-Chimera (*Pettersen et al., 2004*) was used for molecular visualization. The volume of secondary pocket for 251 equally spaced trajectory frames from each run was calculated using POVME 3.0 (*Wagner et al., 2017*). The comparison of volumes between CRY1- CRY2 PHR domain and between CRY2-mutCRY2 PHR domains was done by performing Wilcoxon ranksum test as implemented in Scipy (https://www.scipy.org/). Hydrogen bonds were determined for each frame of the MD trajectories using MDTraj (*McGibbon et al., 2015*). The bonds with at least one residue from the serine loop (residues I36/I54 to I49/I67 in CRY1/CRY2) and with frequency of more than 5000 frames in any trajectory of a particular system were chosen as high frequency hydrogen bonds. The free energy landscape (FEL) was calculated using volume of secondary pocket and distance between center of mass of residue S62 and F123 (S44 and F105 in CRY1), as reaction coordinate. The FEL was calculated by using *gmx sham* utility of Gromacs. For FEL construction the volume and distance were calculated for every $100^{th}$ frame from each trajectory.

## Calculation of Kullback-Leibler (KL) divergence

The KL-divergence or relative entropy has been previously used to quantify the residue wise differences between the ensembles (*McClendon et al., 2012*; *Moffett et al., 2017*; *Rapp et al., 2013*). To calculate the KL-divergence between the ensembles using mutinf software (*McClendon et al., 2012*), we divided each simulation trajectory between 100 ns and 500 ns into 4 blocks of 100 ns each. This gave a total of 12 blocks for apo CRY1 PHR domain and apo CRY2 PHR domain

simulations and eight blocks for the in silico mutCRY2 PHR domain. Dihedral angles phi, psi and chi1 were calculated for 5000 equally spaced frames from each block and used as input to calculate KL-divergence. Dividing the trajectories into multiple blocks of sufficiently large number of frames enabled the calculation of bootstrap values, which were used to make a robust estimation of significant differences between the two ensembles. The divergence values greater than the bootstrap values for a given residue suggests that the dynamics of those residues differ significantly between the two ensembles, in terms of side chain rotamer distribution, main chain dihedral angle distribution or both.

## Crystallization, data collection and structure determination

The CRY1 PHR:PER2 CBD complex was purified as described above. The protein was concentrated to 4.3 mg/mL and crystalized by sitting-drop vapor diffusion at 22 ˚C. Crystals formed in a 1:1 ratio of protein to precipitant in 0.2 M $MgCl_2$, 0.1 M Bis-Tris buffer pH 5.5% and 25% (vol/vol) PEG 3350. Crystals were harvested and flash frozen in the reservoir solution with 20% (vol/vol) glycerol before data collection. Data were collected from single crystal at λ = 1.0A, 100 K on Beamline 23-ID-D at the Advanced Photon Source (Argonne, Illinois, USA). Diffraction images were indexed and scaled using iMosflm (*Battye et al., 2011*) and Scala (*Evans, 2006*) in the CCP4 package (*Winn et al., 2011*). Phases were solved by molecular replacement with PHASER (*Adams et al., 2010*) using the crystal structure of mouse CRY1 PHR domain in complex with the PER2 CBD (PDB: 4CT0). All reflections were used for refinement except for 5% that were excluded for $R_{free}$ calculations. The structural model was revised in real space with the program COOT (*Emsley et al., 2010*) based on sigma-A weighted 2Fo-Fc and Fo-Fc electron density maps. Data collection and final refinement statistics are given in *Supplementary file 1* Table S1.

## Sample preparation for electron microscopy

A complex containing the CLOCK:BMAL1 bHLH-PAS heterodimer bound to an annealed DNA duplex containing the minimal mouse *Per2* E-box (GCG CGG TCA CGT TTT CCA CT) and the CRY1 PHR:PER2 CBD repressor heterodimer was verified from peak fractions of a Superdex 200 10/300 GL gel filtration column (GE Healthcare) and assessed by SDS-PAGE gel electrophoresis and SimplyBlue SafeStain (Invitrogen) staining. The complex was flash frozen and stored at −70 ˚C. For data collection, the complex was briefly incubated on ice in 20 mM HEPES, 125 mM NaCl, 2 mM TCEP and 1% (vol/vol) glycerol. 2.5 µL of sample was then applied to an UltraAuFoil R1.3/1.3 300-mesh grid (Electron Microscopy Services), which was freshly plasma-cleaned using a Gatan Solarus (75% argon/25% oxygen atmosphere, 15 Watts for 7 s). Grids with applied sample were then manually blotted with filter paper (Whatman No.1) for ~3 s in a 4 ˚C cold room before plunge freezing in liquid ethane cooled by liquid nitrogen.

## Electron microscopy data acquisition

The Leginon automated data-acquisition program (*Suloway et al., 2005*) was used to acquire all cryo-EM data. Real-time image pre-processing, consisting of frame alignment, contrast transfer function (CTF) estimation and particle picking, were performed using the Appion image-processing pipeline during data collection (*Voss et al., 2009*). Image collection was performed using a Thermo Fischer Talos Arctica operating at 200 keV and equipped with a Gatan K2 Summit DED, at a nominal magnification of 36,000X (corresponding to a physical pixel size of 1.15 Å per pixel). 692 movies were collected, with 48 frames per movie using a total exposure time of 12 s with an exposure rate of 5.2 e⁻/pixel/s and total exposure of 47.2 e⁻/Å² (0.98 e⁻ per frame). A nominal defocus range from −0.8 µm to −1.8 µm was used during data collection.

## Electron microscopy image processing

Automated particle picking was conducted using the Difference of Gaussians (DoG) picker (*Voss et al., 2009*) to yield 891,700 particle picks. Before particle extraction, CTFFIND4 was used for CTF estimation. Fourier-binned 2 × 2 particles were subjected to reference-free two-dimensional (2D) classification in Relion 3.0 (*Scheres, 2012*) to remove poorly aligning particles and non-particles in the data. A second round of 2D classification in Relion was used to further remove partially denatured or damaged particles and false picks, resulting in 81,257 particles populating 2D classes with

strong structural features that were used for analyses (*Figure 4—figure supplement 2*). Attempts to determine a 3D structure of the entire complex were stymied by a strong preferred orientation of the particles in ice.

## HADDOCK studies

The cluster representatives obtained after clustering the trajectories were used to dock the PAS-B domain (residues 261–395) of CLOCK (PDB: 4F3L) using HADDOCK 2.2 webserver (*van Zundert et al., 2016*). The active residues used for generating restraints for docking were: D38/D56, P39/P57, F41/F59, R51/R69, G106/G124, R109/R127, F257/F275, E382/D400, E383/E401 for CRY1/CRY2 and G332, H360, Q361, W362, E367 for CLOCK PAS-B as before (*Michael et al., 2017a*). The passive residues were specified as residues surrounding the active residues. The PAS-B domain of the CLOCK protein was used to dock an ensemble of structures comprised of 15 cluster representatives of apo CRY1 PHR domain and 12 cluster representatives of apo CRY2 PHR domain. A total of 10000, 400 or 400 models were generated, respectively, for the rigid docking, flexible docking and water refinement phase of the docking protocol. The final 400 models were clustered using Fractional Common Contacts (FCC) and sixteen clusters were obtained comprising 360 structures. Out of these sixteen, there were two large clusters with 92 and 80 complexes constituting about 48% of the clustered structures. The representatives from these two largest clusters were considered for further analysis. Electron microscopy densities were simulated from the atomic models using the 'molmap' function in Chimera, and the densities were low-pass filtered to 8 Å resolution using the 'proc3d' function in EMAN 1.9 (*Ludtke et al., 1999*). Forward projections of each of the two models were generated using the 'project3d' function in EMAN 1.9, and the 'v2' function in EMAN 1.9 was used to compare the projections with the reference-free 2D class averages from RELION.

## Software references

SMOG webtool- http://smog-server.org/.
MDFit- http://smog-server.org/extension/MDfit.html.
Gromacs - http://www.gromacs.org/.

## Acknowledgements

We would like to thank the beamline staff at 23-ID-D for their assistance during data collection at the Advanced Photon Source. We thank JC Ducom at Scripps Research High Performance Computing and C Bowman at Scripps Research for computational support, as well as B Anderson at the Scripps Research Electron Microscopy Facility for microscopy support. We would like to thank Carla Green for the plasmid encoding the mouse *Cry2* 7M mutant. This work was funded by National Institutes of Health (NIH) Grant R01 GM107069 and University of California Cancer Research Coordinating Committee Grant CRN-15–380548 (to CLP), NIH DP2 EB020402 (to GCL), and the RIKEN Dynamic Structural Biology Project (to FT). GCL is supported by the Pew Charitable Trusts as a Pew Scholar and an Amgen Young Investigator award. JLF is supported by the UC Office of the President and a UCSC Chancellor's Postdoctoral Fellowship. CRS was supported by a National Science Foundation predoctoral fellowship. AKM was supported by an NIH NRSA fellowship F31 CA189660. GCGP is supported by an HHMI Gilliam Fellowship with additional support from the UCSC Graduate Division. Computational analyses of EM data were performed using shared instrumentation funded by NIH S10 OD021634 to GCL.

## Additional information

### Funding

| Funder | Grant reference number | Author |
|---|---|---|
| National Institutes of Health | R01 GM107069 | Carrie L Partch |
| Cancer Research Coordinating Committee | CRN-15-380548 | Carrie L Partch |

| | | |
|---|---|---|
| National Institutes of Health | DP2 EB020402 | Gabriel C Lander |
| RIKEN | Dynamic Structural Biology Project | Florence Tama |
| Pew Charitable Trusts | Pew Scholar | Gabriel C Lander |
| Amgen | Young Investigator | Gabriel C Lander |
| University of California | Office of the President Chancellor's Postdoctoral Fellow | Jennifer L Fribourgh |
| National Science Foundation | Graduate Research Fellowship | Colby R Sandate |
| Howard Hughes Medical Institute | Gilliam fellowship | Gian Carlo G Parico |
| National Institutes of Health | F31 CA189660 | Alicia K Michael |
| National Institutes of Health | S10 OD021634 | Gabriel C Lander |

The funders had no role in study design, data collection and interpretation, or the decision to submit the work for publication.

## Author contributions

Jennifer L Fribourgh, Ashutosh Srivastava, Conceptualization, Resources, Formal analysis, Validation, Investigation, Visualization, Writing - original draft, Writing - review and editing; Colby R Sandate, Resources, Formal analysis, Validation, Investigation, Visualization, Writing - original draft, Writing - review and editing; Alicia K Michael, Conceptualization, Resources, Investigation; Peter L Hsu, Christin Rakers, Resources, Investigation; Leslee T Nguyen, Megan R Torgrimson, Gian Carlo G Parico, Resources, Data curation; Sarvind Tripathi, Resources, Validation; Ning Zheng, Conceptualization, Supervision; Gabriel C Lander, Supervision, Funding acquisition, Validation, Visualization, Writing - original draft, Writing - review and editing; Tsuyoshi Hirota, Conceptualization, Supervision, Writing - original draft, Writing - review and editing; Florence Tama, Conceptualization, Supervision, Funding acquisition, Validation, Visualization, Writing - original draft, Writing - review and editing; Carrie L Partch, Conceptualization, Supervision, Funding acquisition, Visualization, Writing - original draft, Project administration, Writing - review and editing

## Author ORCIDs

Ashutosh Srivastava http://orcid.org/0000-0001-9820-720X
Colby R Sandate https://orcid.org/0000-0002-8758-5931
Christin Rakers http://orcid.org/0000-0002-5668-6844
Sarvind Tripathi http://orcid.org/0000-0002-6959-0577
Gabriel C Lander http://orcid.org/0000-0003-4921-1135
Tsuyoshi Hirota http://orcid.org/0000-0003-4876-3608
Carrie L Partch https://orcid.org/0000-0002-4677-2861

## Decision letter and Author response

Decision letter https://doi.org/10.7554/eLife.55275.sa1
Author response https://doi.org/10.7554/eLife.55275.sa2

# Additional files

## Supplementary files

• Supplementary file 1. Details of molecular dynamics simulations. This supplementary file contains two tables (1a, 1b) on separate tabs.

• Supplementary file 2. Details of the experimental structure and computational docking models. This supplementary file contains two tables (2a, 2b) on separate tabs.

• Transparent reporting form

## Data availability

Diffraction data have been deposited in PDB under the accession code 6OF7.

The following dataset was generated:

| Author(s) | Year | Dataset title | Dataset URL | Database and Identifier |
|---|---|---|---|---|
| Michael AK, Fribourgh JL, Tripathi SM, Partch CL | 2019 | Crystal structure of the CRY1-PER2 complex | https://www.rcsb.org/structure/6OF7 | RCSB Protein Data Bank, 6OF7 |

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
