## [Decision Letter]

**Acceptance summary:**

Circadian cycling is a fundamental cellular process that remains poorly understood in molecular terms. By combining insights from many disciplines – X-ray crystallography, electron microscopy, quantitative binding, sizing, and molecular dynamics simulations Fribourgh et al. propose an innovative model for the differing effects and phenotypes associated with the circadian transcriptional repressors, Cry1 and Cry2. Their work shows how another factor, PER2, tunes the binding affinity of CRY1and CRY2 for the CLOCK:BMAL1 transcription factor complex through modulation of a flexible serine-rich loop. They also provide a structure-based model and biochemical evidence for a high-affinity complex comprising CRY1 and CLOCK:BMAL1, in the absence of PER2, which may be the late repressive complex. Together, their data provide explanations for how CRY1 represses BMAL1/CLOCK alone in the late repressive complex. By contrast, CRY2 only suppresses BMAL1/CLOCK when bound to PER2, likely within the early repressive complex.

[Editors' note: the authors submitted for reconsideration following the decision after peer review. What follows is the decision letter after the first round of review.]

**Decision letter after peer review:**

Thank you for submitting your work entitled "Protein dynamics regulate distinct biochemical properties of cryptochromes in mammalian circadian rhythms" for consideration by *eLife*. Your article has been reviewed by three peer reviewers, one of whom is a member of our Board of Reviewing Editors, and the evaluation has been overseen by a Senior Editor. The following individuals involved in the review of your submission have agreed to reveal their identity: Halil Kavakli (Reviewer #3).

Our decision has been reached after consultation between all the reviewers. Based on these discussions and the individual reviews below, we regret to inform you that your work will not be considered for publication in *eLife* at this time. We would like to add, however, that I am open to considering an appeal that addresses the reviewer's comments concerning the functional significance of the advance, the use of and interpretations of the MD simulations, and the concerns about the cryo-EM aspects of your study.

*Reviewer #1:*The manuscript by Fribourgh et al., concerns the structural basis of circadian rhythm regulation by two paralogous repressors of transcription, CRY1 and CRY2, and their modulator, PER2. Each CRY protein binds to the PAS-B domain of CLOCK, which together with BMAL1, forms the heterodimeric transcription factor complex that cycles "on" (when free of CRY1 and CRY2) and "off" (when repressed by CRY1 or CRY2). Notably, CRY1 and CRY2 knockout mice have different phenotypes, with CRY2 being insufficient for ~24hour circadian cycling, despite high sequence conservation. This study attempts to explain how these repressor proteins differ in their structural dynamics and therefore differentially regulate circadian cycling.

The authors start by showing that CRY1 binds to CLOCK:BMAL(bHLH-PAS) heterodimers with an ~18 fold tighter affinity than CRY2 (K_d_ values of ~65nM versus ~1.2uM). Comparison of different structural models (some published, some based on MD modeling, and some generated from a new crystal form in the current manuscript) suggest a plausible explanation for CRY1 versus CRY2 differential binding affinity. In short, CRY1 and CRY2 vary in sequence in the critical "serine loop" and the nearby PAS domain-binding "secondary pocket." These sequence differences alter the size and inferred dynamics of the secondary pocket. Consistently, seven key substitutions in this region of CRY2 make that make it more like CRY1 confer tighter binding to CLOCK:BMAL(bHLH-PAS) – which is expected given published data concerning the CRY2 7M mutant.

Second, Fribourgh et al., investigate how a critical modulatory protein, PER2, binds and remodels the structure of the serine loop and the secondary binding pocket for CRY1 and CRY2. A comparison of their new crystal structure of CRY1:PER2(CBD) with prior structures (including PDBs: 6OF7,4I6E) revealed a surprise. Inclusion of PER2 in the complex partially orders the serine loop, and this correlates with weakened binding between PER2:CRY1 and CLOCK:BMAL(bHLH-PAS), by comparison with CRY1 alone. By contrast, PER2 binding to CRY2 leads to more extensive remodeling of the serine loop and secondary pocket, and this correlates with ~2 fold tighter binding between CRY2 and CLOCK:BMAL(bHLH-PAS). Synthesizing these observations, the authors conclude that PER2(CBD) can "tune" the affinities of CRY1 and CRY2 for CLOCK:BMAL1(bHLH-PAS) by restructuring the serine loop, and that this activity largely normalizes the ternary complex stabilities (collapsing a ~18-fold difference for the PER2-free complexes to a ~3 fold difference for the PER2-containing complexes).

Overall, this study has some crisp new insights into PER2's effects on CRY1 versus CRY2 structure and binding properties, but it remains unclear to me how this work should change our thinking about circadian rhythmicity since, as the authors wrote, after accounting for PER2's opposing effects on CRY1 versus CRY2, "the actions of CRY1 and CRY2 within the heteromultimeric PER-CRY repressive complexes may be largely similar." Is the central takeaway that the ~3 fold binding difference for CRY2, in the presence of PER2, the mechanistic basis for CRY2's limitations as a suppressor? Finally, the cryo-EM and associated structural modeling efforts based on the cryo-EM suffer from limitations that make it of unclear value to the manuscript.

Essential revisions:

1) The title of the paper seems too generic to me, bordering on insignificant since protein dynamics regulate biochemistry in all contexts, not just circadian rhythms. What mechanistic take-home message about the CRY1/2 and PER proteins would the authors suggest as an *eLife* title that conveys the significance of their discovery?

2) I had some trouble navigating the diverse depictions of the structures. If Figure 1A defines the standard view, please add clear labels for the serine loop, the secondary pocket, the footprint of the PAS domain, and the imprint of the TAD domain here and then use this figure with standard rotation symbols to help the reader understand which surface features are being focused on and highlighted in subsequent figures.

3) MD simulations are thought-provoking, fine-grained structural hypotheses, but I do not think of them as experimental data. I am happy to see such models in the supplemental data, but it's not clear to me how useful they were in thinking about the differences between CRY1 and CRY2, or the consequences of PER2 binding for either CRY proteins. For example, I may be confused by Figure 3 suggests that models of CRY2=>CRY1 mutations (the 7M mutant) display the strongest effects for residues far away from the serine loop (near aa250 and aa300), while the serine loop and especially the secondary pocket (around aa400) are less impressively altered in RMSF.

4) The authors use unnecessarily polemical language when describing a prior structure as "non-physiological" and an "artifact". Yes, vector residues are a problem, but all crystal structures have limitations. I would encourage the authors to describe their work as an attempt to glean explanatory power from all of the available structural data-which when considered together suggests considerable flexibility and scope for regulatory remodeling in this region of CRY1/2.

5) The authors wrote that they attempted to resolve outstanding ambiguity about the structure of PER2(CBD)CRY1(PHR):CLOCK:BMAL1(bHLH-PAS)+dsDNA using single particle cryoEM. I am afraid the uncertainty remains, and the cryo-EM data and associated analysis are both incomplete and disconnected from the rest of the study. Despite claiming a resolution between ~6Ã… and ~7Ã…, I can only appreciate a few apparent Î±-helices in the figures. For the most part, the map lacks secondary structure element separation, and this precludes a reliable interpretation of their map. The unbiased class averages shown in Figure 4—figure supplement 2B bear only a partial resemblance to projections of the author's pseudoatomic model. So, while their low-resolution fitting exercise may be a reasonable guess, I cannot endorse this structural model based on the cryo-EM data as presented.

*Reviewer #2:*This is a follow-on paper to the authors' earlier work which established the structure of a repressive complex involving the circadian clock proteins CRY1, CLOCK, and BMAL1. The present study seeks to understand how it is that CRY1 and CRY2 have similar structures yet function differently. The authors show experimentally that CRY1 and CRY2 have similar binding affinities for BMAL1 but different binding affinities for CLOCK; the presence of PER2 stabilizes the CRY2:CLOCK:BMAL1 complex two-fold. Based on molecular dynamics simulations, they attribute the differences in affinity to differences in flexibility of a loop adjacent to the binding pocket for CLOCK (dubbed the serine loop). This loop is unstructured (or at least largely unresolved crystallographically) in CRY1. By contrast, in CRY2, in the absence of PER2, it forms an Î± helix and, in the presence of PER2, takes on an alternate structure consistent with data for the loop in CRY1. This is an interesting result since it seems to contradict the intuition that a more well-structured complex will be tighter. Along the way, they solve a structure of the CRY1:PER2 complex that removes a vector artifact and model the CRY1:CLOCK complex based on cryoEM and biophysical data.

1) It would be nice if the authors could better explain the forces that stabilize the different serine loop conformations. Given that they use molecular dynamics simulations to generate hypotheses, they could use restrained free energy perturbation simulations to identify the interactions favoring different conformations, for example. Or try to map the (free) energy landscapes more systematically.

2) This is potentially a study of interest to readers beyond those working on circadian clocks since overall differences in function of related proteins (e.g., isoforms) are not well understood, at least quantitatively. In this regard, the manuscript could be written more accessibly for readers less familiar with the clock.

*Reviewer #3:*In the present study Fribourgh at al., use a combination of structural biology and computational methods to describe association of CRY1 and CRY2 with CLOCK:BMAL1. They found that a highly dynamic Ser looping and size of secondary packet in the CRYs is responsible for their affinity to CLOCK:BMAL1 complex. They specifically found that the binding of PER2 to CRY2 is required for its repressor activity due remodeling of Ser loop by PER2. However, they showed CRY1 can directly interact with PAS domain of CLOCK with or without PER2 and, therefore strongly repress CLOCK:BMAL1 transactivation. In this aspect this study is important and provides biochemical evidence how both CRYs have differential repressor activity. This topic is potentially appropriate for *eLife*, followings should be address in the manuscript before publication.

1) Experiments in Figure 1C and Figure 2 should have done with a type of negative control like using photolyase or Drosphila CRY (since it has small cavity in their secondary packet) to validate their results.

2) Repression assay (commonly done in the field) should be performed using appropriate cell line to support their conclusion indicated in the Results section.

3) It is hard believe whether CRY2 binds to C:B bHLH-PAS since peak of CRY2 PHR: C:B bHLH-PAS and C:B bHLH-PAS overlaps Figure 2B. For validation some other CRY type from plant/insect in both experiments. Alternatively, trimeric complex needs to be shown by either SDS PAGE (as shown in Figure 2—figure supplement 1) or Western blot.

4) Subsection "The serine loop differentially gates access to the secondary pocket of CRY1 and CRY2": Change in RMSF is not clearly shown in Figure 3D for A61G. Although they cited papers (mutants that used in these papers are different than CRY2 A61G) they should confirm their findings using CRY2-A61G by measuring the mutant repressor capacity and its effect on circadian rhythm using MEF double knockout CRY cell line.

5) Subsection "The serine loop differentially gates access to the secondary pocket of CRY1 and CRY2" "these data demonstrate that modest sequences differences..."

I disagree with this statement. CRY2 7M consists of seven mutations in and around secondary packet and such number of mutations can make things complicated.

---

## [Author Response]

[Editors' note: the authors resubmitted a revised version of the paper for consideration. What follows is the authors' response to the first round of review.]

Reviewer #1:The manuscript by Fribourgh et al., concerns the structural basis of circadian rhythm regulation by two paralogous repressors of transcription, CRY1 and CRY2, and their modulator, PER2. [...] Finally, the cryo-EM and associated structural modeling efforts based on the cryo-EM suffer from limitations that make it of unclear value to the manuscript.

We thank reviewer #1 for their thorough and careful review of our manuscript. We agree that this study provides insight into the outstanding question of how PER2 influences the affinity of cryptochromes for CLOCK:BMAL1. We believe that localizing the molecular basis of this differential affinity to the serine loop in the PHR domain of CRYs represents an important advance in our understanding of the molecular circadian clock. However, another key finding that perhaps wasn't appreciated by the reviewer is our discovery that CRY1 has a very high affinity for the PAS domain core of CLOCK:BMAL1, which provides a molecular basis to understand the prior observation from ChIP-seq studies that CRY1 can act alone as a repressor of CLOCK:BMAL1, late in the circadian cycle. We too agree that the quality of the cryo-EM data were not as high as we would have hoped for, given issues with interdomain flexibility of CLOCK:BMAL1 and the severe preferred orientation of the complex in ice. We describe this in more detail below, but we believe that we have addressed reviewer #1's concern with the limitations of the cryo-EM data by removing the 3D model and instead using 2D class averages to validate a computational model of the ternary PER2:CRY1:CLOCK complex.

Altogether, we believe that our study gives rise to 3 discoveries that are important for circadian rhythms: (1) by showing that differential repressive functions of CRY1 and 2 likely don't originate from their affinity for the BMAL1 TAD, but the PAS domain core of CLOCK:BMAL1, we highlight a critical protein-protein interface that is likely to be regulated to control clock timing; (2) our discovery that PER2 acts to equalize CRY1 and CRY2 affinity for CLOCK:BMAL1 through modulation of the serine loop provides a structure-based rationale for the assembly and function of CRY1/2-containing early repressive complexes, and (3) our discovery of the very high affinity that CRY1 has for CLOCK:BMAL1 in the absence of PER2 provides a foundation to understand its important but enigmatic role in the late repressive complex.

Essential revisions:1) The title of the paper seems too generic to me, bordering on insignificant since protein dynamics regulate biochemistry in all contexts, not just circadian rhythms. What mechanistic take-home message about the CRY1/2 and PER proteins would the authors suggest as an eLife title that conveys the significance of their discovery?

We acknowledge that our original title was too broad and didn't fully describe the scope of the study. We have changed the title to "Dynamics at the serine loop underlie differential affinity of cryptochromes for CLOCK:BMAL1 to control circadian timing."

2) I had some trouble navigating the diverse depictions of the structures. If Figure 1A defines the standard view, please add clear labels for the serine loop, the secondary pocket, the footprint of the PAS domain, and the imprint of the TAD domain here and then use this figure with standard rotation symbols to help the reader understand which surface features are being focused on and highlighted in subsequent figures.

Thanks for the suggestion. We defined a standard view of the protein in Figure 1A and use this with proper rotation symbols in subsequent figures to orient the reader. We also altered our labeling strategy to decrease confusion. We believe that this should improve the ability to navigate the structural depictions presented in the manuscript.

3) MD simulations are thought-provoking, fine-grained structural hypotheses, but I do not think of them as experimental data. I am happy to see such models in the supplemental data, but it's not clear to me how useful they were in thinking about the differences between CRY1 and CRY2, or the consequences of PER2 binding for either CRY proteins. For example, I may be confused by Figure 3 suggests that models of CRY2=>CRY1 mutations (the 7M mutant) display the strongest effects for residues far away from the serine loop (near aa250 and aa300), while the serine loop and especially the secondary pocket (around aa400) are less impressively altered in RMSF.

We appreciate the reviewer's opinion that MD simulations can be thought-provoking and politely encourage the reviewer to consider how extensively they have been used to make important contributions exploring the dynamics and function of biomolecules. We feel that the MD simulations make very important contributions to our understanding of serine loop dynamics in the cryptochromes and should remain in the main figures.

With regard to the question of how useful they were in thinking about the differences between CRY1 and CRY2, we believe that the MD simulations served a critical role here, by complementing our experimental binding data and providing a molecular mechanism linking the dynamics of the serine loop near the secondary pocket to the affinity of cryptochromes for CLOCK:BMAL1. The static crystal structures of CRY1 and CRY2 only suggested that differences in the flexibility exist at the serine loop; therefore, there was no clarity about the factors that might lead to differential flexibility and whether this has a role in the functional difference observed between CRY1 and CRY2. To address this comment (and one from reviewer #2), we added new analyses of the MD simulations to demonstrate that swapping in two residues from CRY1 into the serine loop of CRY2 (i.e., mutCRY2) not only recapitulates CRY1-like flexibility in the loop, but also its pattern of hydrogen-bonding and the free energy landscape. Furthermore, we added a biochemical characterization of the MD-inspired mutant CRY2 2M to demonstrate that sequence differences in the loop that alter its dynamics do modulate CRY affinity for CLOCK:BMAL1.

With respect to the example chosen by the reviewer for RMSF differences, we highlight that proteins are dynamic in nature and MD simulations do an excellent job of capturing these dynamics computationally. The regions highlighted by reviewer are indeed both flexible loops that show considerable conformational heterogeneity with high B-factors in the crystal structures of all CRYs. Even in the simulation trajectories, the RMSF values show a large standard deviation for these residues. On the other hand, the serine loop and the secondary pocket in apo CRY2 is primarily composed of secondary structures like a-helices and b-strands, generally considered to be relatively rigid. Thus, we believe that the increase in their flexibility due to two mutations in the serine loop is functionally relevant, regardless of the magnitude in comparison to other long, flexible loops.

4) The authors use unnecessarily polemical language when describing a prior structure as "non-physiological" and an "artifact". Yes, vector residues are a problem, but all crystal structures have limitations. I would encourage the authors to describe their work as an attempt to glean explanatory power from all of the available structural data-which when considered together suggests considerable flexibility and scope for regulatory remodeling in this region of CRY1/2.

We completely agree with the reviewer here––we weren't trying to use overly contentious rhetoric, but we believe that it is imperative to note that the vector sequence included in the PER2 construct used in the prior structure provided an incorrect depiction of the structure of the serine loop in the CRY1:PER2 complex. We agree that softening the language here is merited, so we modified language in this section and moved a panel from the original figure (clearly labeling the artifact) to a supplemental figure.

5) The authors wrote that they attempted to resolve outstanding ambiguity about the structure of PER2(CBD)CRY1(PHR):CLOCK:BMAL1(bHLH-PAS)+dsDNA using single particle cryoEM. I am afraid the uncertainty remains, and the cryo-EM data and associated analysis are both incomplete and disconnected from the rest of the study. Despite claiming a resolution between ~6Ã… and ~7Ã…, I can only appreciate a few apparent Î±-helices in the figures. For the most part, the map lacks secondary structure element separation, and this precludes a reliable interpretation of their map. The unbiased class averages shown in Figure 4—figure supplement 2B bear only a partial resemblance to projections of the author's pseudoatomic model. So, while their low-resolution fitting exercise may be a reasonable guess, I cannot endorse this structural model based on the cryo-EM data as presented.

We agree with reviewer #1 about the unsatisfyingly low resolution of our single particle cryo-EM studies. In our original manuscript we did note that the resolution estimate reported by Relion is likely inflated by preferred orientation: "Some anisotropy was observed in our cryo-EM density, arising from particles adopting two preferred orientations in vitrified ice (Figure S6), which was detrimental to the resolution of our 3D reconstruction and likely leads to an inflated value for our final map. This is likely arising from interaction with the air-water interface with our complex, as commonly seen in other protein samples analyzed by cryo-EM..."

We were frustrated with the dynamic nature of the multidomain complex (documented earlier in small-angle x-ray scattering studies, Michael et al., (2017)) and with the several preferred orientations of the complex that we fought with throughout our EM studies. In the end, we felt that having some experimental data validating the binding of CLOCK PAS-B into the secondary pocket of a CRY1:PER2 complex was worth including in the initial draft of the manuscript. We acknowledge that the low resolution of our 3D map leaves some ambiguity in structural interpretation of the 3D model, so we have removed the 3D map from this study, instead placing more value on the 2D class averages arising from the same dataset. We note that in contrast to our 3D map, these 2D class averages contain well-resolved secondary structural features for the CRY1 PHR and are less likely to suffer from the same ambiguity in interpretation. Additionally, we have included a more expansive comparison between the 2D classes and the forward projections of the two top-scoring pseudoatomic HADDOCK models (Figure 4—figure supplement 2B). Critically, the EM data were able to clearly support one of the two models to provide a better picture of the ternary complex between PER2, CRY1, and CLOCK. However, we respectfully disagree with reviewer #1 in their assessment that our EM analyses are disconnected from the overall results of the study. To date, there is still no high resolution structure available describing the interactions between the mammalian circadian transcription factors and their repressors, giving our model considerable value to the circadian community. To that end, the EM data were instrumental in selecting the most probable structural model obtained from the integrative MD and HADDOCK results. We have added new text in subsection "The CRY1 serine loop is disordered in the CRY1 PHR:PER2 CBD complex" outlining this new work.

Reviewer #2:1) It would be nice if the authors could better explain the forces that stabilize the different serine loop conformations. Given that they use molecular dynamics simulations to generate hypotheses, they could use restrained free energy perturbation simulations to identify the interactions favoring different conformations, for example. Or try to map the (free) energy landscapes more systematically.

We agree that a discussion about the interactions stabilizing different conformations of the serine loop would be a valuable addition to the manuscript. However, performing restrained free energy perturbation simulations would be computationally expensive and we believe out of scope for the current manuscript. In order to more systematically explore the serine loop conformations as per the reviewer's suggestion, we have now plotted Free Energy Landscapes (FEL) corresponding to volume and distance at the entrance of the secondary pocket. The FEL clearly shows a difference in the relative distribution of frames with respect to the volume and distances among CRY1, CRY2 and mutCRY2 (Figure 3—figure supplement 1D). We also extracted the frames corresponding to the minimum in each FEL and compared their volume and distance. These again showed the same trend as the average values mentioned in the original manuscript with mutCRY2 behaving similar to the apo CRY1 (Supplementary file 1).

Regarding the characterization of interactions stabilizing the serine loop, it is difficult to comment here on the PER2 interactions since the simulations here were performed for apo CRY1 and CRY2. We agree that this is an interesting path forward in understanding the dynamics of this loop and its interaction with PER2 and CLOCK PAS-B. This is a part of continued work that we are pursuing for another manuscript. However, taking the reviewer's suggestion into account, we have analyzed the hydrogen bonds formed by secondary pocket loop residues over the course of the trajectories described in the manuscript and added new text describing the work in subsection "The serine loop differentially gates access to the secondary pocket of CRY1 and CRY2" and in Supplementary file 2. We feel that this analysis adds an important depth to the study and thank the reviewer for the suggestion to pursue this line of study.

2) This is potentially a study of interest to readers beyond those working on circadian clocks since overall differences in function of related proteins (e.g., isoforms) are not well understood, at least quantitatively. In this regard, the manuscript could be written more accessibly for readers less familiar with the clock.

We agree with the reviewer that the fundamental findings of this study could be of interest to protein aficionados beyond the circadian clock field. We have made some modest alterations to text throughout the manuscript to make it more accessible to nonclock readers. However, at its core, this study addresses an important question about the function of the molecular circadian clock and we feel that it is necessary to frame our findings in light of the transcription/translational oscillator instead of more general principles of protein behavior. Focusing on this general aspect of our study could be a great idea for a future review on this topic.

Reviewer #3:1) Experiments in Figure 1C and Figure 2 should have done with a type of negative control like using photolyase or Drosphila CRY (since it has small cavity in their secondary packet) to validate their results.

We agree with the reviewer that negative controls are of the utmost importance. In this case, the negative controls for these assays have already been performed elsewhere and these studies are appropriately cited. We respectfully disagree that using a known non-repressive homolog like *E. coli* photolyase or *Drosophila* CRY would be the best control in this assay. The negative controls for the fluorescence polarization (FP)-based TAD binding assay in Figure 1C were originally published in Xu et al., (2015) , where we used TAD mutants that reduced repression by CRY in 293T cells and also showed decreased affinity for CRY by both FP and ITC, two gold-standard methods for determining affinity; moreover, the affinities that we measure here are in line with the data we published earlier. The negative controls for CRY binding to the PAS domain core of CLOCK:BMAL1 were published in Michael et al., (2017), where we showed that the W362A point mutant of CLOCK significantly disrupts or eliminates altogether the interaction of CLOCK with CRY1 by size exclusion chromatography and in vitro pulldown assays. Both of these citations are made in the text when describing prior knowledge in this area, so we believe that this should address the Reviewer's primary concern.

2) Repression assay (commonly done in the field) should be performed using appropriate cell line to support their conclusion indicated in the Results section.

The ability of CRY1 and CRY2 to repress CLOCK:BMAL1 independently of PER2 was published many years ago in Shearman et al., (2000), using transfection into the heterologous S2-based insect cell system. Moreover, this observation has been repeated in numerous papers, some of which explored the nature of repression when CRYs were expressed alone or with PER2 (e.g., Akashi et al., (2014)). The emphasis in this concluding sentence was meant to summarize our current findings that both CRY proteins bind the TAD similarly (with or without PER2). To connect this claim to the observation that sequestration of the TAD by CRYs is likely the model by which they directly repress CLOCK:BMAL1 activity, we have added a citation to our earlier work where we showed that CRY1 and the KIX domain of the coactivator CBP directly compete for the same binding site on the BMAL1 TAD (e.g., in Xu et al., (2015)). We believe that linking the summary of our current findings to this prior study that rigorously explored TAD binding with in vitro and cellular repression studies should serve as proper support for this claim.

3) It is hard believe whether CRY2 binds to C:B bHLH-PAS since peak of CRY2 PHR: C:B bHLH-PAS and C:B bHLH-PAS overlaps Figure 2B. For validation some other CRY type from plant/insect in both experiments. Alternatively, trimeric complex needs to be shown by either SDS PAGE (as shown in Figure 2—figure supplement 1) or Western blot.

We acknowledge the point that the reviewer makes, although we point out that binding of the bHLH-PAS heterodimer to the similarly sized CRY1 PHR domain significantly shifts the elution profile of the complex and there is no reason to think that the CRY2 PHR would behave substantively different in this assay. However, to clear up any potential confusion, we have added SDS-PAGE gel from both chromatography runs (with volume markers) to Figure 2—figure supplement 1, which clearly shows that the chromatographic peaks for CRY2 and the CLOCK:BMAL1 heterodimer do not overlap. We respectfully disagree with the reviewer that we need to run a negative control using a non-repressive CRY from plants or insects here, as the data are quite clear that CRY2 does not stably migrate with CLOCK:BMAL1. We also probed for complex formation of CRY2 with the bHLH-PAS heterodimer using size exclusion chromatography in line with multi-angle scattering (SEC-MALS), which provides shape-independent determination of mass to unambiguously validate complex formation. However, in line with the SEC data presented here, we only identified masses for the isolated CLOCK:BMAL1 heterodimer and CRY2 PHR domain, and not a complex of the two (data not shown). Therefore, we believe that addition of the SDS-PAGE in Figure 2—figure supplement 1 should satisfy the reviewer's concern that we perhaps missed stable complex formation between the CRY2 PHR and the bHLH-PAS heterodimer.

4) Subsection "The serine loop differentially gates access to the secondary pocket of CRY1 and CRY2": Change in RMSF is not clearly shown in Figure 3D for A61G. Although they cited papers (mutants that used in these papers are different than CRY2 A61G) they should confirm their findings using CRY2-A61G by measuring the mutant repressor capacity and its effect on circadian rhythm using MEF double knockout CRY cell line.

The Reviewer makes a good point that mutants used in the Rosensweig et al., (2017) are, in some cases, a bit different from the ones probed in our assays here. However, this is only partially correct, as we purposefully studied the CRY2 7M mutant biochemically because it had already been validated in rigorous cellular studies by Rosensweig et al. In this manuscript, we added experimental data to bolster the MD simulations of mutCRY2, probing a CRY2 PHR with the A61G/N64S mutations that substitutes two residues from CRY1 into CRY2 to destabilize the serine loop dynamics. Rosensweig et al., studied the converse mutant in their cellular reconstitution studies––they made the G43A/N46S mutant in CRY1 to show that it weakened repression and shortened circadian period, in line with their model that this mutant had a weaker, more CRY2-like repression phenotype. Our logic for using the CRY2 2M mutant (instead of making the corresponding CRY1 mutant like Rosensweig et al.,) was that it would allow for a direct comparison to the affinity we obtained with the CRY2 7M mutant. We have added text in this draft ( subsection "The serine loop differentially gates access to the secondary pocket of CRY1 and CRY2".) to explain this connection. In this regard, we believe that the data with the CRY2 2M and 7M mutations have been properly supported by citations.

5) Subsection "The serine loop differentially gates access to the secondary pocket of CRY1 and CRY2" "these data demonstrate that modest sequences differences...."I disagree with this statement. CRY2 7M consists of seven mutations in and around secondary packet and such number of mutations can make things complicated.

We agree with the reviewer that seven mutations does not seem like a modest number, but we were referring to the actual amino acid substitutions themselves, in which residues of similar size and/or chemical character were substituted for one another (i.e., K/R, E/D, W/F, etc.). However, we acknowledge that this characterization is unclear, so we have removed the word 'modest' from our description of the 7M mutant in subsection "The serine loop differentially gates access to the secondary pocket of CRY1 and CRY2".